# Adaptive Transformer Programs: Bridging the Gap Between Performance and Interpretability in Transformers

**Quoc-Vinh Lai-Dang**[1], **Taemin Kang**[2], **Seungah Son**[1]
[1]Cho Chun Shik Graduate School of Mobility    [2]The Robotics Program
Korea Advanced Institute of Science and Technology (KAIST)
Daejeon, South Korea
{ldqvinh, tmkang9826, seungahson}@kaist.ac.kr

## Abstract

Balancing high performance with interpretability in increasingly powerful Transformer-based models remains a challenge. While mechanistic interpretability aims to specify neural network computations in explicit, pseudocode-like formats, existing methods often involve laborious manual analysis or struggle to fully elucidate learned internal algorithms. Recent efforts to build intrinsically interpretable models have introduced considerable expressivity and optimization challenges. This work introduces *Adaptive Transformer Programs*, an enhanced framework building upon RASP language and Transformer Programs to create more robust and interpretable models. The proposed method increases expressivity by redesigning two primary attention modules to improve categorical and numerical reasoning capabilities. To overcome optimization hurdles, we introduce a novel reparameterization scheme that enhances the exploration-exploitation trade-off during training. We validate our approach through extensive experiments on diverse tasks, including in-context learning, algorithmic problems (e.g., sorting and Dyck languages), and NLP benchmarks such as named entity recognition and text classification. Results demonstrate that Adaptive Transformer Programs substantially narrow the performance gap between black-box Transformers and interpretable models, enhancing transparency. This work advances the development of high-performing, transparent AI systems for critical applications, addressing crucial ethical concerns in AI development.

## 1 Introduction

Balancing high performance with model interpretability has emerged as a central challenge in artificial intelligence. The introduction of Transformer architectures (Vaswani et al., 2017) and the rise of large language models (LLMs) (Brown et al., 2020) have significantly advanced natural language processing. However, these powerful models often operate as "black boxes," making it difficult to understand their decision-making processes. This issue is especially critical in fields like healthcare, finance, and law, where AI-driven decisions can have profound impacts. Addressing this challenge within the context of Transformers and LLMs is both timely and essential.

Various interpretability techniques have been proposed to illuminate how AI models make decisions, each offering unique insights yet presenting distinct challenges. Behavioral approaches, such as those by Ribeiro et al. (2020); Warstadt et al. (2020), probe model responses to diverse inputs, providing an external view of model behavior but lacking access to internal reasoning mechanisms. Attribution methods like Integrated Gradients (Sundararajan et al., 2017) and SmoothGrad (Smilkov et al., 2017) quantify the influence of input features on predictions but often fail to capture underlying causal relationships. Concept-based interpretabilities (Kim et al., 2018; Belinkov, 2022) adopt a top-down approach to unraveling a model's decision-making processes but risk introducing biases through subjective concept selection. Mechanistic interpretability efforts (Elhage et al., 2021; Nanda et al., 2023) delve into the internal computations of models but struggle with scalability as model

complexity grows. These limitations underscore the necessity for inherently interpretable models that offer transparent decision-making processes without sacrificing performance.

Advancements such as RASP, Tracr, and Transformer Programs represent significant strides toward inherently interpretable Transformer models. RASP (Weiss et al., 2021) introduces a programming language that allows users to define Transformer operations in a human-readable format, effectively mapping neural computations to symbolic logic. Building on this, Tracr (Lindner et al., 2024) serves as a compiler that translates RASP programs into actual Transformer weights, bridging the gap between high-level specifications and low-level implementations. Transformer Programs (Friedman et al., 2024) take this a step further by proposing a method to train Transformers that can be directly translated into discrete, interpretable programs. While these innovations move us closer to transparent AI systems, challenges in expressivity and optimization persist. Our work addresses these challenges by introducing novel enhancements to Transformer Programs.

In this paper, we introduce three key innovations that enhance the expressivity and optimization of Transformer Programs (Friedman et al., 2024) while preserving interpretability. First, we propose a seamless transition mechanism between Gumbel-Softmax and Sparsemax, improving the exploration-exploitation trade-off during training by allowing the model to dynamically adjust its attention distributions. Second, we develop an uncertainty-aware attention mechanism that integrates categorical and score-based attention through Jensen-Shannon Divergence, enabling the model to handle varying levels of uncertainty in data processing. Third, we enhance the numerical mechanism by incorporating positional encodings. These contributions not only extend the functional capacity of Transformer Programs but also maintain their inherent interpretability, addressing limitations in previous approaches.

Our extensive validation on diverse tasks, including in-context learning, algorithmic problems (Weiss et al., 2021), and NLP benchmarks, demonstrates the effectiveness of our *Adaptive Transformer Programs*. Experimental results show a substantial improvement in bridging the performance gap between black-box Transformers and interpretable models while offering enhanced transparency. This work not only advances the state-of-the-art in interpretable AI but also paves the way for the responsible and ethical integration of AI systems in critical applications, potentially transforming how we develop and deploy AI in high-stakes environments.

## 2 BACKGROUND

**Transformer Architecture and Circuits.** The Transformer architecture (Vaswani et al., 2017) has revolutionized sequential data processing, achieving unprecedented performance across NLP tasks. It processes token sequences $\boldsymbol{w} = \{w_1, w_2, \ldots, w_N\}$ from a vocabulary $\mathcal{V}$, converting each into a high-dimensional embedding. The initial representation $\mathbf{x}_0 \in \mathbb{R}^{N \times d}$ combines learned token embeddings with positional encodings, crucial for capturing sequential information. The architecture consists of $L$ layers, each refining the input representation through two main components: Multi-Head Attention (MHA) and Multilayer Perceptron (MLP). The output of layer $i$ is computed as:

$$\mathbf{x}_i = \mathbf{x}_{i-1} + \text{MLP}_i(\mathbf{x}_{i-1} + \text{MHA}_i(\mathbf{x}_{i-1})), \tag{1}$$

where MHA allows the model to attend to different positions within the sequence:

$$\text{MHA}(\mathbf{x}) = \sum_{h=1}^{H} \text{softmax}\left(\frac{\mathbf{x}\boldsymbol{W}_Q^h(\mathbf{x}\boldsymbol{W}_K^h)^\top}{\sqrt{d_k}}\right)\mathbf{x}\boldsymbol{W}_V^h\boldsymbol{W}_O^h. \tag{2}$$

Recent research has focused on understanding Transformers through the lens of "Transformer circuits" (Elhage et al., 2021), viewing them as a residual stream architecture where each component reads from and writes to a running representation:

$$\mathbf{x}_i = \mathbf{x}_{i-1} + f(\mathbf{x}_{i-1}\boldsymbol{W}_{\text{in}})\boldsymbol{W}_{\text{out}} \tag{3}$$

This approach has yielded insights into emergent behaviors, such as induction heads for in-context learning (Olsson et al., 2022). While offering insights into attention heads and neurons, their ap-

proach to interpretability is limited by the complexity of modern models. This has led to efforts to bridge the gap between symbolic reasoning and neural network models.

**Bridging Transformers and Programs.**  Bridging the gap between Transformers and symbolic programs has emerged as a promising direction for enhancing interpretability. RASP (Weiss et al., 2021) offers a programming language designed to express Transformer computations in a human-readable format. Its key `select` function, analogous to attention in Transformers, which takes sequences of keys $k \in \mathcal{K}^N$ and queries $q \in \mathcal{Q}^M$, along with a boolean predicate $p : \mathcal{Q} \times \mathcal{K} \to \{0, 1\}$, to produce an attention matrix $A \in \{0, 1\}^{M \times N}$. This is followed by an `aggregate` operation, akin to value aggregation in Transformers. Building on RASP, Lindner et al. (2024) introduced Tracr, a compiler that converts RASP programs into Transformer weights. This led to Learning Transformer Programs (Friedman et al., 2024), a method for training Transformers that can be automatically converted into discrete, human-readable programs. This approach advances neural-symbolic integration by combining neural network expressiveness with symbolic transparency. However, a key challenge remains in implementing effective discrete optimization to ensure both accuracy and interpretability in the learned programs.

**Discrete Optimization.**  Discrete optimization is crucial for training interpretable models with discrete representations like Transformer Programs. The Gumbel-Softmax estimator (Jang et al., 2017) enables differentiable sampling from discrete distributions, generating one-hot encoded vectors approximating discrete selections. This allows gradient-based optimization in Transformers. While effective, it has limitations in finding optimal solutions and promoting sparsity, crucial for interpretability. This paper explores alternative methods, including Sparsemax (Martins & Astudillo, 2016), and introduces a novel smooth transition mechanism addressing these limitations, leading to more effective, interpretable program learning.

## 3 ADAPTIVE TRANSFORMER PROGRAMS

### 3.1 OVERVIEW

Our approach builds upon the Transformer Programs framework (Friedman et al., 2024), which introduces two key constraints for interpretable Transformers: a disentangled residual stream and rule-based modules.

The disentangled residual stream encodes each program variable in a dedicated, orthogonal subspace, preventing the entanglement often seen in standard Transformers (Vaswani et al., 2017) and facilitating clear reading and writing mechanisms. When reading, each module accesses specific variables using projection matrices parameterized by one-hot indicator vectors. Formally, if the residual stream encodes $m$ categorical variables, each with cardinality $k$, resulting in input embeddings $\mathbf{x} \in \{0, 1\}^{N \times mk}$, then each projection matrix $\mathbf{W} \in \mathbb{R}^{mk \times k}$ is defined by an indicator $\boldsymbol{\pi} \in \{0, 1\}^m$: $\mathbf{W} = [\pi_1 \mathbf{I}_k; \dots; \pi_m \mathbf{I}_k]^\top$, where $\mathbf{I}_k$ is the $k \times k$ identity matrix. Writing involves concatenating new information to maintain separation: $\mathbf{x}_i = [\mathbf{x}_{i-1}; h(\mathbf{x}_{i-1})]$, where $i$ denotes the layer and $h$ is an attention head.

Transformer Programs enforce interpretable, rule-based mappings between input and output variables. Categorical attention heads compute attention patterns using boolean predicate matrices and employ hard attention for aggregation. The attention pattern is determined using a boolean predicate matrix $\mathbf{W}_{\text{predicate}} \in \{0, 1\}^{k \times k}$, defining mappings between query and key values. This results in an attention score matrix $\mathbf{S} \in \{0, 1\}^{N \times N}$ where $\mathbf{S} = \mathbf{x} \mathbf{W}_Q \mathbf{W}_{\text{predicate}} (\mathbf{x} \mathbf{W}_K)^\top$. Hard attention ensures each query attends to a single key, producing a categorical output variable: $\mathbf{A}_i = \text{One-hot}\left(\arg\max_j \mathbf{S}_{i,j}\right)$. Additional modules include factored categorical embeddings, limited numerical attention, and feed-forward layers as lookup tables.

While effective, the original framework's use of Gumbel-Softmax reparameterization (Jang et al., 2017) faces challenges in finding optimal solutions and promoting sparsity. Our work addresses these challenges through three main contributions: (1) a Smooth Transition Mechanism for discrete optimization, (2) Uncertainty-Aware Categorical Attention, and (3) Position-Aware Numerical Attention. These enhancements improve both interpretability and performance, leading to *Adaptive Transformer Programs*.

## 3.2 SMOOTH TRANSITION MECHANISM FOR DISCRETE OPTIMIZATION

Our Smooth Transition Mechanism facilitates effective discrete optimization in Transformer Programs (Friedman et al., 2024) by gradually shifting from exploration to exploitation during training. The inherently discrete nature of Transformer Programs, with modules containing discrete parameters like predicate matrices and gate vectors, poses challenges for traditional gradient-based methods. To address this, differentiable relaxation techniques have been employed, with the Gumbel-Softmax estimator (Jang et al., 2017) being widely adopted:

$$\tilde{z}_i = \frac{z_i + g_i}{\tau} \tag{4}$$

$$y_{\text{soft},i} = \text{softmax}_i(\tilde{\mathbf{z}}) = \frac{\exp(\tilde{z}_i)}{\sum_j \exp(\tilde{z}_j)}. \tag{5}$$

where $g_i$ is Gumbel noise, $\tau$ is temperature, $z_i$ is raw logit, and $\tilde{z}_i$ is perturbed and scaled logit. However, Gumbel-Softmax (Jang et al., 2017) often yields sub-optimal programs due to local optima and fails to encourage sparsity, hindering interpretability and efficiency.

To address these limitations, we introduce a Smooth Transition Mechanism combining Gumbel-Softmax and Sparsemax (Martins & Astudillo, 2016). This hybrid approach balances exploration and exploitation during training. Initially, it behaves like Gumbel-Softmax, encouraging diverse program structures. As training progresses, it shifts towards a Sparsemax variant with Gumbel noise, which we term Gumbel-Sparsemax:

$$y_{\text{sparse},i} = \text{sparsemax}(\tilde{\mathbf{z}}) := \underset{\mathbf{p} \in \Delta^{K-1}}{\arg\min} \|\mathbf{p} - \tilde{\mathbf{z}}\|^2. \tag{6}$$

where $\Delta^{K-1}$ is the $(K-1)$-dimensional probability simplex, and the equation finds the closest point to the perturbed logit $\tilde{\mathbf{z}}$. This promotes sparsity and more deterministic program choices, refining promising solutions and encouraging concise, interpretable programs. This balance enables the discovery of high-quality, interpretable, and efficient program structures.

The temperature parameter $\tau$ controls the smooth transition between Gumbel-Softmax and Gumbel-Sparsemax. High $\tau$ favors exploration (Gumbel-Softmax), while low $\tau$ promotes exploitation (Gumbel-Sparsemax). The transition is governed by $\alpha(\tau)$:

$$\alpha(\tau) = \frac{\tau_1 - \tau}{\tau_1 - \tau_2} \tag{7}$$

where $\tau_1, \tau_2$ are the transition points ($\tau_1 > \tau_2$). The hybrid distribution is:

$$y = (1 - \alpha(\tau)) \cdot y_{\text{soft}} + \alpha(\tau) \cdot y_{\text{sparse}} \tag{8}$$

While improving discrete optimization, this mechanism falls short of fully addressing the need for adaptability and robustness in real-world scenarios, motivating our next contribution: Uncertainty-Aware Attention.

## 3.3 UNCERTAINTY-AWARE ATTENTION

Categorical attention heads in Transformer Programs, as in Tracr (Lindner et al., 2024) and RASP (Weiss et al., 2021), enforce one-to-one attention, excelling at discrete, rule-based relationships but struggling with nuanced or continuous relationships. Weiss et al. (2021) proposed an extension to RASP combining a binary function $\text{predicate} : \mathcal{Q} \times \mathcal{K} \to \{0, 1\}$ with a continuous function $\text{score} : \mathcal{Q} \times \mathcal{K} \to \mathbb{R}$, capturing fine-grained relationships useful for tasks like semantic similarity in NLP. This continuous score function forms the basis of what we term *score-based attention*. While Friedman et al. (2024) suggested incorporating score-based attention in Transformer Programs, this approach increases program complexity. Our preliminary experiments with separate

binary (predicate-based) and continuous functions (score-based) revealed scenario-dependent performance variations, motivating the development of a hybrid mechanism to create more Adaptive Transformer Programs.

We employ Jensen-Shannon Divergence (JSD) (Lin, 1991), a symmetric and smoothed version of Kullback-Leibler divergence (Kullback & Leibler, 1951), to measure uncertainty in categorical attention. JSD's non-negativity, boundedness, and symmetry make it suitable for this task. In this context, JSD measures uncertainty in probability distributions, where higher values indicate greater uncertainty and a larger divergence between categorical attention and a reference distribution. Given a query $q$ and keys $k_1, \ldots, k_n$, we define categorical attention (CatAttention) as $\mathbf{A}_{\text{cat},i} = \text{predicate}(q, k_i)$ and score-based attention (ScoreAttention) as $\mathbf{A}_{\text{score},i} = \text{score}(q, k_i)$. We introduce a dynamic reference attention $\mathbf{A}_{\text{ref},i}$ that adapts during training, allowing flexible uncertainty estimation. The JSD is formulated as:

$$\text{JSD}(\mathbf{A}_{\text{cat},i} \parallel \mathbf{A}_{\text{ref},i}) = \frac{1}{2}\text{KL}(\mathbf{A}_{\text{cat},i} \parallel \mathbf{A}_{\text{avg},i}) + \frac{1}{2}\text{KL}(\mathbf{A}_{\text{ref},i} \parallel \mathbf{A}_{\text{avg},i}) \tag{9}$$

where $\mathbf{A}_{\text{avg},i} = (\mathbf{A}_{\text{cat},i} + \mathbf{A}_{\text{ref},i})/2$. This formulation enables uncertainty estimation in the attention mechanism, which is used to adjust attention weights accordingly.

A learnable gating mechanism, driven by the JSD-based uncertainty estimate, dynamically weights the contributions of CatAttention and ScoreAttention. The gating mechanism: $g = \text{MLP}(\text{JSD}(\mathbf{A}_{\text{cat},i} \parallel \mathbf{A}_{\text{ref},i}))$, that implemented as a network module that takes the JSD value as input and outputs a gating weight between 0 and 1. This weight is used to combine the outputs of CatAttention and ScoreAttention:

$$\mathbf{A}_i = g \cdot \mathbf{A}_{\text{cat},i} + (1 - g) \cdot \mathbf{A}_{\text{score},i} \tag{10}$$

In high uncertainty scenarios (high JSD), the gate favors ScoreAttention, which is more reliable in uncertain contexts. In low uncertainty (low JSD), CatAttention is preferred, as it is more confident in its categorical decisions. This adaptive mechanism provides flexible and robust decision-making by dynamically adjusting the balance between attention types based on uncertainty.

While Uncertainty-Aware Attention improves the handling of categorical and contextual information, processing numerical data poses additional challenges. To address this, we introduce the Position-Aware Attention module, which extends the original Numerical Attention mechanism.

### 3.4 POSITION-AWARE ATTENTION

The numerical attention mechanism in Transformer Programs (Friedman et al., 2024) is restricted to outputting integer values within a bounded range. This limitation hinders the model's ability to represent and process continuous or fractional values, thereby reducing expressiveness and complicating tasks that require nuanced numerical representations or complex calculations. Additionally, numerical attention employs a binary predicate matrix and computes a weighted sum instead of a weighted average, simplifying standard Transformer attention and diminishing the model's capacity to capture intricate relationships between inputs. Furthermore, numerical variables are limited to being either constant (set to one at the input layer) or outputs of numerical attention heads, constraining the model's ability to learn and represent arbitrary numerical values and restricting its problem-solving capabilities.

Our Position-Aware Attention mechanism extends the numerical attention in Transformer Programs (Friedman et al., 2024). It uses categorical variables as keys and queries, and numerical variables as values. We incorporate both Learnable (Gehring et al., 2017) and Sinusoidal (Vaswani et al., 2017) Positional Encodings into the numerical value variable `var`, creating a position-aware value `var_pos`. This allows the model to learn nuanced positional relationships. Given attention scores $\mathbf{S} \in \{0, 1\}^{N \times N}$, the output for the $i^{th}$ token is computed as:

$$\mathbf{A}_{\text{num},i} = \sum_{j=1}^{N} S_{i,j} \texttt{var}_{\text{pos}}[j] \tag{11}$$

The integration of learnable and sinusoidal positional encodings offers several advantages. Learnable encodings capture nuanced positional information and enable processing of non-integer values, expanding the model's numerical capabilities for tasks requiring fine-grained representations. By incorporating positional information, each token can distinguish between identical numerical values at different positions. Additionally, Sinusoidal Positional Encodings provide a structured approach to embedding positional information, improving the model's ability to handle sequence-dependent tasks and maintain interpretability. These enhancements, combined with the Smooth Transition Mechanism and Uncertainty-Aware Attention, enable Transformer Programs to be effectively converted into interpretable programs, as discussed in the subsequent section on Experimental Results.

Table 1: Accuracy (Acc.) and Program Length (Lines) for Transformer Programs (Baseline) and Adaptive Transformer Programs (Ours) on In-Context Learning (Friedman et al., 2024) and RASP tasks Weiss et al. (2021).

| Dataset | Description | Example | Baseline | | Ours | |
|---------|-------------|---------|----------|------|------|------|
| | | | Acc. | Lines | Acc. | Lines |
| Induction | In-context learning. | `induction("a1b2b2a") = 1` | 100.0 | 107 | 100.0 | 101 |
| Reverse | Reverse the order. | `reverse("abbc") = "cbba"` | 99.74 | 859 | 99.99 | 779 |
| Histogram | Count the number of tokens. | `hist("abbc") = "1221"` | 99.94 | 199 | 99.95 | 189 |
| Double hist. | Count the number of unique tokens sharing identical frequency count. | `hist2("abbc") = "2112"` | 66.78 | 586 | 91.81 | 513 |
| Sort | Arrange the input elements in alphabetically ascending order. | `sort("cbba") = "abbc"` | 99.98 | 945 | 99.86 | 895 |
| Most-Freq | Order unique elements by occurrence frequency, using earlier positions to break ties. | `most_freq("abbc") = "bac"` | 76.44 | 1334 | 80.80 | 894 |
| Dyck-1 | Classify if each position $i$ is a valid string(T), a valid prefix(P), or an invalid(F). | `dyck1("(())") = "PTPTF"` | 99.69 | 1297 | 99.93 | 1086 |
| Dyck-2 | The same analysis with above, but in Dyck-2. | `dyck2("()[])") = "PPPPTPF"` | 97.98 | 1316 | 98.14 | 1065 |

## 4 EXPERIMENTS

### 4.1 EXPERIMENTAL SETUP AND DATASETS

To evaluate the effectiveness of Adaptive Transformer Programs, we conducted experiments on a diverse set of tasks: a simple in-context learning task (Friedman et al., 2024), algorithmic RASP tasks (Weiss et al., 2021), and two standard NLP tasks—named entity recognition using CoNLL-2003 (Tjong Kim Sang & De Meulder, 2003) and text classification using TREC, MR, Subj, and AG News datasets (Voorhees & Tice, 2000; Pang & Lee, 2004; 2005; Zhang et al., 2015). In the in-context learning task, the model processed sequences of up to 10 tokens of alternating letters and numbers from a vocabulary of four letters and four numbers, outputting the number following a repeated letter or *unk* for a new letter, using an attention-only Transformer with two layers and one attention head per layer, fixed one-hot encoded token and position variables as input, and a causal attention mask. For the RASP tasks, as summarized in Table 1, we tested our models on small-scale datasets with sequence lengths up to 16 (Dyck tasks) or 8 (others), using vocabularies matching the sequence lengths; the models employed fixed one-hot token and position embeddings, variable cardinality set to the maximum sequence length, and incorporated our enhanced modules—Uncertainty-Aware categorical attention and Position-Aware numerical attention heads and MLPs with two input variables. In the NLP tasks, sentences were limited to 32 words for named entity recognition and 64 words for text classification, both using a 10,000-word vocabulary and initialized with 300-dimensional GloVe embeddings (Pennington et al., 2014); for named entity recognition, only categorical attention heads and MLPs were employed, while text classification used averaged token embeddings for sentence representation.

### 4.2 IN-CONTEXT LEARNING AND RASP TASKS

**Performance.** Table 1 compares Adaptive Transformer Programs (Ours) to Transformer Programs (Friedman et al., 2024) (Baseline) across eight tasks, showing improvements in accuracy and program complexity. Our approach consistently matches or outperforms the baseline, with notable gains

in challenging tasks: Double histogram (91.81% vs 66.78%) and Most-Freq (80.80% vs 76.44%). For tasks like Induction and Sort, where baseline accuracy was already high, our model maintains performance (100% and 99.86% respectively) while reducing program complexity. Slight accuracy improvements are observed in Reverse, Histogram, and Dyck-1 tasks. Interestingly, for the Dyck-2 task, which involves more complex string analysis, our model achieves a small increase in accuracy from 97.98% to 98.14%. These results demonstrate that Adaptive Transformer Programs maintain high accuracy across diverse RASP tasks while showing significant improvements in challenging scenarios, highlighting the effectiveness of our proposed enhancements in complex reasoning tasks.

**Interpretability.** A key advantage of Adaptive Transformer Programs is their ability to achieve more concise and interpretable representations, evidenced by the reduced program length (lines of code) across all tasks in Table 1. This improved sparsity not only suggests greater computational efficiency but also enhances interpretability, making it easier to understand the model's decision-making process. Notably, our approach achieves substantial reductions in program length for both complex tasks like Dyck-1 (16.3% reduction) and Dyck-2 (19.1% reduction) and simpler tasks like Induction (5.6% reduction) and Histogram (5% reduction). The most significant reduction is observed in the Most-Freq task (33% reduction, from 1334 to 894 lines). These improvements in conciseness, combined with maintained or improved accuracy across tasks, demonstrate the effectiveness of Adaptive Transformer Programs in balancing performance with interpretability.

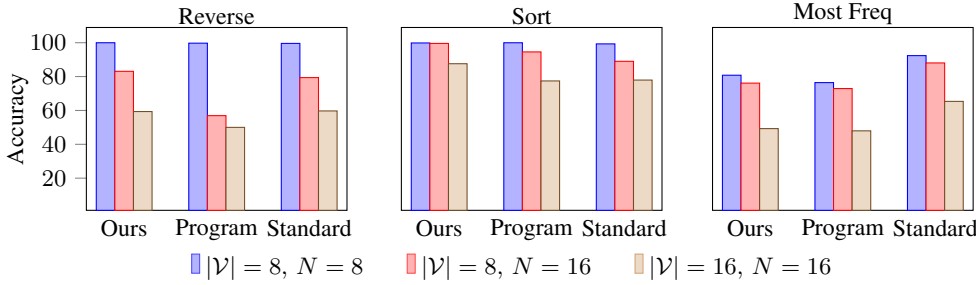

Figure 1: RASP accuracy comparison of Adaptive Transformer Programs (Ours), Program Transformers (Program), and Standard Transformers (Standard) across increasing input vocabulary sizes ($|\mathcal{V}|$) and sequence lengths ($N$).

**Scalability.** Adaptive Transformer Programs demonstrate robust scalability when faced with increasing input complexity, outperforming both Program Transformers and Standard Transformers in some scenarios. Figure 1 illustrates the performance across three representative RASP tasks (Reverse, Sort, and Most Freq) as we increase the input vocabulary size ($|\mathcal{V}|$) from 8 to 16 and the maximum sequence length ($N$) from 8 to 16. For the Reverse task, our model maintains high accuracy (99.99%) with $|\mathcal{V}| = 8$ and $N = 8$, and experiences less degradation (83.11% and 59.35%) compared to baselines as complexity increases. In the Sort task, Adaptive Transformer Programs consistently outperform Standard Transformers and show better resilience than Program Transformers, maintaining 87.6% accuracy even at $|\mathcal{V}| = 16$ and $N = 16$. The Most Freq task presents a challenge for all models, but our approach still demonstrates competitive performance, particularly at higher complexities. These results highlight the superior scalability of Adaptive Transformer Programs, showing their potential for handling more complex, real-world data distributions while maintaining interpretability.

**Ablation Study.** Table 2 presents our comprehensive ablation study, evaluating the impact of three key enhancements: the Smooth Transition Mechanism, Uncertainty-Aware Attention, and Position-Aware Attention on the Most-Freq task. We manually enable or disable these components in various combinations, training separate models for each configuration. The results reveal that the full model, with all enhancements enabled, achieves the highest accuracy of 80.8% while maintaining a relatively concise program length of 894 lines. As observed in the ablation study where only Smooth Transition is enabled and the others disabled, while the Smooth Transition Mechanism (which incorporates Gumbel-Sparsemax) contributes significantly to program conciseness, it also shows a slight decrease in accuracy when used in isolation, highlighting the trade-off between interpretability and

Table 2: Ablation study on Most-Freq task: Impact of model enhancements on performance (Accuracy) and program length (Lines).

| Smooth Transition Mechanism | Uncertainty-Aware Attention | Position-Aware Attention | Accuracy | Lines |
|:---:|:---:|:---:|:---:|:---:|
| ✓ | ✓ | ✓ | 80.8 | 894 |
| ✓ | ✓ | - | 78.25 | 850 |
| ✓ | - | ✓ | 75.01 | 939 |
| - | ✓ | ✓ | 78.62 | 938 |
| ✓ | - | - | 73.61 | 920 |
| - | ✓ | - | 77.34 | 896 |
| - | - | ✓ | 76.26 | 1028 |
| - | - | - | 76.44 | 1334 |

performance. Disabling individual components leads to performance drops, with Uncertainty-Aware Attention showing the most significant impact (accuracy decrease to 75.01% when disabled). Notably, removing all enhancements (equivalent to baseline Transformer Programs) results in a lower accuracy (76.44%) and the longest program (1334 lines), highlighting the cumulative benefit of our proposed enhancements. When multiple components are disabled, the performance declines further—removing both Uncertainty-Aware Attention and Position-Aware Attention leads to an accuracy of 73.61% and a less efficient program length of 920 lines. These results demonstrate the critical role each enhancement plays in maintaining high performance and concise, interpretable program structures.

## 4.3 NLP Tasks

Table 3: NER Performance Metrics and Program Length on CoNLL-2003 Dataset.

| Model | Accuracy | Precision | Recall | F1 | Lines |
|:---|:---:|:---:|:---:|:---:|:---:|
| Standard Transformers | 92.2 | 71.1 | 62.5 | 66.6 | - |
| Transformer Programs | 94.2 | 78.9 | 72.9 | 75.8 | 991 |
| Ours | 94.1 | 77.2 | 73.2 | 75.1 | 916 |

**Named Entity Recognition (NER).** On the CoNLL-2003 Named Entity Recognition (NER) task, a standard benchmark for sequence labeling, Adaptive Transformer Programs demonstrate competitive performance while offering significant advantages in interpretability. Table 3 presents the results, comparing our approach to Standard Transformers and Transformer Programs. Our model achieves 94.1% accuracy, closely matching the 94.2% of Transformer Programs and significantly outperforming the 92.2% of Standard Transformers. Although the F1 scores are similar across Transformer Program and our approach (75.8 and 75.1, respectively), Adaptive Transformer Programs achieve this performance with a notably shorter program length (916 lines compared to 991). This conciseness, indicative of greater program sparsity, is a key advantage, promoting easier analysis and understanding of the learned programs, a crucial aspect for interpretability. This result highlights the ability of Adaptive Transformer Programs to maintain competitive performance while generating more interpretable program representations.

Table 4: Accuracy and Program Length for Various Text Classification Tasks.

| Model | TREC | | MR | | Subj | | AG | |
|:---|:---:|:---:|:---:|:---:|:---:|:---:|:---:|:---:|
| | Acc. | Lines | Acc. | Lines | Acc. | Lines | Acc. | Lines |
| Standard Transformer | 83.4 | - | 75.9 | - | 90.9 | - | 89.1 | - |
| Transformer Program | 84.2 | 5520 | 77.1 | 3972 | 92.3 | 3065 | 90.3 | 1881 |
| Ours | 83.6 | 827 | 77.9 | 773 | 90.4 | 1954 | 90.0 | 1790 |

**Text Classification.** Adaptive Transformer Programs exhibit robust performance across diverse text classification tasks, demonstrating their capacity for generalization to various real-world sce-

narios. As shown in Table 4, our approach performs competitively on four distinct classification tasks: TREC (question type identification), MR (sentiment evaluation), Subj (subjectivity assessment), and AG news (topic categorization). Notably, our model achieves the highest accuracy on the MR task at 77.9%, surpassing both Standard Transformers (75.9%) and Transformer Programs (77.1%). While performance on other tasks is comparable to the baselines, with slight variations in accuracy, the most striking difference lies in program length. Across all tasks, Adaptive Transformer Programs consistently produce more concise programs. For instance, on the TREC task, our model requires only 827 lines compared to 5520 lines for Transformer Programs, representing an 85% reduction in complexity. This significant decrease in program length, coupled with competitive accuracy, underscores the efficiency and interpretability of our approach in handling diverse text classification challenges.

## 5 RELATED WORK

**Learning Programs.** Program synthesis has evolved from classical symbolic approaches to deep learning-based methods, driven by the need to scale to complex programs learned by modern neural architectures. Traditional paradigms like Inductive Logic Programming (Muggleton & de Raedt, 1994) and Deductive Program Synthesis (Manna & Waldinger, 1980) relied on symbolic reasoning and expert knowledge. The field then shifted towards neural program induction, with works like Neural Programmer-Interpreters (Reed & de Freitas, 2016) and Neuro-Symbolic Program Synthesis (Devlin et al., 2017) learning programs directly from data. However, these methods struggle with scalability to large datasets, complex program structures, and incorporating domain-specific knowledge (Gulwani et al., 2017). Transformer Programs (Friedman et al., 2024) address these limitations by leveraging Transformer architectures' representation learning capabilities while imposing constraints to learn interpretable programs.

**Transformers and Formal Languages.** Recent research has demonstrated the expressive power of Transformers in relation to formal languages. Studies show that Transformers can learn regular and context-free languages, and implement algorithms like first-order logic with majority quantifiers (Hahn, 2020; Merrill & Sabharwal, 2022). Work by Giannou et al. (2023) further supports the view of Transformers as general-purpose computation devices. Weiss et al. (2021) established an initial connection between Transformer operations and program-like representations through the RASP language. Adaptive Transformer Programs build on this foundation, enhancing interpretability and programmatic representation to align Transformers with human-understandable symbolic systems.

**Interpretable Machine Learning Models.** The field of interpretable machine learning has seen a surge in methods for understanding deep learning models. Post-hoc methods include attention visualization (Bahdanau et al., 2014), feature attribution (Ribeiro et al., 2016; Lundberg & Lee, 2017), and concept activation vectors (Kim et al., 2018). Architectural modifications, such as sparse attention (Zhang et al., 2021) and inductive biases (Geiger et al., 2024), attempt to enhance interpretability through model design. In contrast, intrinsically interpretable models offer direct access to underlying algorithms, improved transparency, and the potential for formal verification. Our Adaptive Transformer Programs aim to learn inherently interpretable models, providing more faithful and complete explanations of decision-making processes. This approach addresses the limitations of post-hoc methods and architectural modifications by representing complex computations transparently.

**Uncertainty in Deep Learning.** Uncertainty estimation plays a crucial role in developing reliable and interpretable deep learning models. Quantifying uncertainty improves model reliability (Kendall & Gal, 2017), facilitates human-AI collaboration (Gal & Ghahramani, 2016), and enhances interpretability (Leibig et al., 2017). Prominent techniques include Bayesian Neural Networks (MacKay, 1995), Monte Carlo Dropout (Gal & Ghahramani, 2016), and Ensemble Methods (Lakshminarayanan et al., 2017). Our Uncertainty-Aware Attention mechanism dynamically combines attention types based on uncertainty estimates, leading to more robust and interpretable models. This approach uniquely integrates program synthesis, Transformer architectures, interpretability, and uncertainty estimation.

# 6 CONCLUSION AND DISCUSSION

The study presents a novel framework for developing robust, expressive, and interpretable Transformer models that can be translated into human-readable programs. Key contributions include a Smooth Transition Mechanism for discrete optimization, an Uncertainty-Aware Attention mechanism for adaptive attention blending, and a Position-Aware Attention module for numerical reasoning. Empirical results on synthetic and real-world NLP tasks demonstrate superior performance compared to benchmarks and provide concise, insightful interpretability analysis. This work advances interpretable AI by improving performance and clarity through new adaptive mechanisms, facilitating the creation of transparent and reliable AI systems. Additionally, it explores the convergence of program synthesis, deep learning, and uncertainty estimation, promoting accountability and the societal benefits of AI.

**Integration of Contributions.** The three enhancements introduced in this work synergistically contribute to the effectiveness of Adaptive Transformer Programs. The Smooth Transition Mechanism promotes program sparsity by gradually shifting from exploration-focused Gumbel-Softmax to exploitation-focused Gumbel-Sparsemax. Uncertainty-Aware Attention dynamically adapts the attention strategy based on uncertainty estimates, enhancing expressiveness and robustness. Position-Aware Attention improves numerical reasoning and training stability through positional encodings.

**Future Research Directions.** Adaptive Transformer Programs exhibit potential but face challenges such as scaling to larger models and tasks, necessitating improved training methods or compact representations. The complexity of these programs can impede human understanding, highlighting the need for simplification, summarization, or visualization techniques. Extending their application to computer vision or robotics will require adapting knowledge representation and extraction processes. Future research might explore advanced structures like recursion and hierarchical composition for greater expressiveness and leverage interpretability to support human-AI collaboration through interactive program refinement tools.

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

# A  PROGRAM EXTRACTION AND INTERPRETATION

## A.1  DETAILED EXTRACTION PROCESS

### A.1.1  DEFINITION OF "LINES OF CODE" METRIC

To quantify the complexity and interpretability of extracted programs, we define "lines of code" (LOC) as the number of active program components derived from a trained Transformer model. The LOC metric serves as a quantitative measure of program complexity and interpretability. In our framework, LOC is formally defined as the sum of active rules or operations in the symbolically extracted program representation. Specifically, LOC is calculated by summing the contributions from the following program components:

- **Predicate Functions:** For each predicate function, we count the number of conditional branches (e.g., `if`, `elif`, `else`) and the `return` statement. Each such statement constitutes one line of code.

- **Attention Pattern Selection:** Each invocation of functions like `select_closest` or `select` to determine attention patterns is counted as one line. These functions represent the core attention mechanism.

- **Aggregation Functions:** Each call to aggregation functions such as `aggregate` or `aggregate_sum`, which combine attention outputs, contributes one line to the LOC.

- **Output Score Calculation:** The operations involved in accessing and processing the `classifier_weights` to compute output scores are also counted. Typically, indexing operations and subsequent processing contribute to the LOC.

### A.1.2  PROGRAM EXTRACTION METHODOLOGY

Our program extraction methodology translates learned model parameters into symbolic code, adapting to the Gumbel-Sparsemax transition and uncertainty-aware attention mechanisms. The key steps are detailed below:

**Gumbel-Sparsemax Transition.** The Smooth Transition Mechanism, employing a blend of Gumbel-Softmax and Sparsemax, primarily influences the *training process* and the resulting *learned parameters*. It does not necessitate changes to the fundamental *program extraction methodology* itself. The extraction process for predicate matrices, attention weights, and aggregation operations remains consistent with the original Transformer Programs. The effect of the Gumbel-Sparsemax transition is observed indirectly in the extracted programs: the learned predicates tend to be sparser, and the attention patterns are more focused. These characteristics, which contribute to program simplicity and reduced "lines of code," are a *consequence* of the improved optimization facilitated by the Smooth Transition Mechanism, rather than a direct change in the extraction procedure.

**Uncertainty-Aware Attention.** The Uncertainty-Aware Attention mechanism introduces a dynamic blending of categorical and score-based attention, guided by Jensen-Shannon Divergence (JSD)-based gating. While the extraction process for the individual components (categorical predicates, score functions, and aggregation) remains fundamentally similar, the key adaptation lies in how we interpret the extracted program and understand the influence of uncertainty awareness.

The core idea is that uncertainty-aware training shapes the *learned parameters* of the categorical predicates and score functions to be more robust and contextually appropriate. During extraction, we observe the *resulting predicate logic and attention patterns* which reflect the effects of this uncertainty-aware training. For instance, due to uncertainty-aware gating:

- **More Generalized Predicates:** Extracted predicates may exhibit greater generalization, being less sensitive to minor input variations, reflecting the influence of score-based similarity under high uncertainty.

- **Context-Aware Grouping:** The program may demonstrate groupings of query positions with similar behaviors, even if their categorical features are not identical, indicating learning driven by contextual similarity guided by uncertainty.

In summary, the extraction process for ATP retains the core principles of the original Transformer Programs method. The novelty lies in the enhanced *training mechanisms* that lead to *qualitatively different and more interpretable extracted programs*. The "lines of code" metric effectively captures the resulting reduction in program complexity stemming from these mechanisms.

### A.1.3 IMPACT OF UNCERTAINTY-AWARE ATTENTION ON PROGRAM STRUCTURE

While the extraction methodology remains largely consistent, Uncertainty-Aware Attention significantly impacts the *structure* and *complexity* of the extracted programs. This mechanism allows the model to adapt its attention strategy dynamically based on uncertainty, leading to more robust and efficient program logic. During training, this dynamic gating, driven by the JSD-based uncertainty estimate, encourages the model to:

- **Learn Robust Predicates:** By blending with score-based attention when categorical attention is uncertain (high JSD), the model learns categorical predicates that are less brittle and more generalized. This reduces the need for highly specific and numerous conditional clauses within predicates, thus simplifying predicate definitions and decreasing LOC.

- **Develop Context-Appropriate Attention:** The adaptive blending allows the model to utilize categorical attention when confident (low JSD) and rely more on score-based attention in ambiguous or noisy contexts (high JSD). This context-dependent attention strategy leads to more efficient information processing and potentially reduces the complexity of subsequent program modules.

- **Achieve Concise Program Logic:** The overall effect of learning more robust predicates and context-appropriate attention is a more streamlined and concise program structure. This manifests as fewer "lines of code" due to simplified predicates, more efficient module interactions, and reduced redundancy in program logic.

In essence, Uncertainty-Aware Attention guides the learning process to favor program structures that are not only accurate but also more robust and interpretable by dynamically adapting attention strategies based on the context-dependent uncertainty.

### A.1.4 DIFFERENCES IN "LINES OF CODE" ACROSS TRAINED MODELS

The "lines of code" metric can vary significantly between trained models, even for the same task, reflecting differences in the learned program structures. This variability arises from several factors:

- **Optimization Path and Randomness:** Neural network training involves stochastic optimization. Different random initializations and training trajectories can lead to models converging to different, yet functionally similar, solutions. Some solutions may be more concise and interpretable (lower LOC) than others due to variations in the optimization path taken.

- **Effectiveness of Learning Mechanisms:** The effectiveness of mechanisms like Gumbel-Sparsemax and Uncertainty-Aware Attention in guiding the model towards simpler and more efficient solutions directly influences the final program length. Models trained with ATP, leveraging these mechanisms, tend to learn programs with lower LOC compared to baseline models.

- **Underlying Program Structure Efficiency:** Different models might discover different algorithmic approaches to solve the same task. Some algorithms are inherently more complex (requiring more steps and rules) than others. ATP aims to facilitate the discovery of more efficient, algorithmically simpler solutions, leading to reduced LOC.

### A.2 EXAMPLE PROGRAM COMPARISONS

This section presents side-by-side comparisons of extracted programs for the Double Histogram and Histogram tasks, demonstrating the improved conciseness and interpretability achieved by Adaptive Transformer Programs (ATP) compared to baseline Transformer Programs. We analyze the specific program structures and quantify the reduction in "lines of code" (LOC) and accuracy improvements.

A.2.1  DOUBLE HISTOGRAM TASK COMPARISON

We compare code snippets from Layer 2, Attention Head 0 (`attn_1_0`) and Attention Head 1 (`attn_1_1`) for both baseline and ATP models trained on the Double Histogram task.

**Baseline Transformer Program**

```
# Baseline code: attn_1_0
def predicate_1_0(q_position, k_position):
    if q_position in {0}:
        return k_position == 7
    elif q_position in {1}:
        return k_position == 2
    elif q_position in {2}:
        return k_position == 3
    elif q_position in {3}:
        return k_position == 5
    elif q_position in {4, 5}:
        return k_position == 6
    elif q_position in {6}:
        return k_position == 1
    elif q_position in {7}:
        return k_position == 4
attn_1_0_pattern = select_closest(positions, positions, predicate_1_0)
```

**Adaptive Transformer Program (ATP)**

```
# Our code: attn_1_0
def predicate_1_0(q_position, k_position):
    # Grouped positions: {0, 3, 4} mapped to k_position == 7
    if q_position in {0, 3, 4}:
        return k_position == 7
    # Grouped positions: {1, 2, 5} mapped to k_position == 6
    elif q_position in {1, 2, 5}:
        return k_position == 6
    elif q_position in {6}:
        return k_position == 1
    elif q_position in {7}:
        return k_position == 0
attn_1_0_pattern = select_closest(positions, positions, predicate_1_0)
```

Figure 2: Side-by-Side Comparison of `attn_1_0` Predicates for Double Histogram Task. The ATP program groups query positions (e.g., $\{0, 3, 4\}$ and $\{1, 2, 5\}$) to improve efficiency.

**Analysis of `attn_1_0` Comparison (Figure 2):**

* **Grouped Query Positions:** The ATP program groups query positions within the `if` and `elif` conditions in `predicate_1_0`, as highlighted in `q_position in {0, 3, 4}` and `q_position in {1, 2, 5}`. The baseline program lacks this grouping, having separate conditions for each query position (e.g., `elif q_position in {0}:`, `elif q_position in {3}:`, `elif q_position in {4}:` in a fully expanded version).

* **Concise Predicate Logic:** Grouping query positions significantly reduces the number of conditional checks, leading to more concise logic in ATP's predicate. ATP's `predicate_1_0` has fewer `elif` clauses compared to a fully expanded baseline predicate.

* **Line Count Reduction:**

  • Baseline `predicate_1_0` (expanded): $\approx$ **9 lines** (if fully expanded based on separate conditions)

- ATP `predicate_1_0`: **7 lines**

**Baseline Transformer Program**

```
# Baseline code: attn_1_1
def predicate_1_1(q_position, k_position):
   if q_position in {0}:
       return k_position == 6
   elif q_position in {1, 3}:
       return k_position == 4
   elif q_position in {2}:
       return k_position == 7
   elif q_position in {4}:
       return k_position == 2
   elif q_position in {5}:
       return k_position == 3
   elif q_position in {6, 7}:
       return k_position == 5
attn_1_1_pattern = select_closest(positions, positions, predicate_1_1)
```

**Adaptive Transformer Program (ATP)**

```
# Our code: attn_1_1
def predicate_1_1(attn_0_0_output, position):
   # Conditions on attn_0_0_output
   if attn_0_0_output in {0}: # Condition on prev. output
       return position == 4
   elif attn_0_0_output in {1, 5, 7}: # Condition on prev. output
       return position == 6
   elif attn_0_0_output in {2, 3, 4}: # Condition on prev. output
       return position == 5
attn_1_1_pattern = select_closest(positions, attn_0_0_outputs, predicate_1_1)
```

Figure 3: Side-by-Side Comparison of `attn_1_1` Predicates for Double Histogram Task. Note that ATP uses conditions based on `attn_0_0_output`.

**Analysis of `attn_1_1` Comparison (Figure 3):**

\* **Conditioning on Previous Attention Output:** ATP's `predicate_1_1` conditions on `attn_0_0_output` (output of the previous attention head) instead of just `position` as in the baseline. This is highlighted in `if attn_0_0_output in {0}:`. This indicates a more hierarchical and efficient logic, where attention decisions are informed by the outputs of earlier processing stages.

\* **Compact Logic via Output Conditioning:** By conditioning on the output of `attn_0_0`, ATP achieves a significantly more compact `predicate_1_1` with fewer clauses. The baseline predicate requires more specific positional rules, whereas ATP leverages the information processed by `attn_0_0`.

\* **Line Count Reduction:**

- Baseline `predicate_1_1`: **9 lines**
- ATP `predicate_1_1`: **7 lines**

**Overall Impact on Double Histogram Task:**

The combined effect of grouped query positions in `attn_1_0` and output-conditioned logic in `attn_1_1` in ATP leads to a substantial reduction in program length and improved accuracy for the Double Histogram task:

* **Line Count Reduction: 586 lines (Baseline) → 513 lines (ATP)**

* **Accuracy Improvement: 66.78% (Baseline) → 91.81% (ATP)**

This example clearly demonstrates how ATP learns more concise and efficient programs by leveraging its novel mechanisms. The grouped query positions and hierarchical, output-conditioned logic contribute to both improved interpretability (shorter program, easier to understand logic) and enhanced performance.

# B    METHOD DETAILS AND ANALYSIS

## B.1    SMOOTH TRANSITION MECHANISM

### B.1.1    VISUALIZATION OF $\alpha(\tau)$ EVOLUTION OVER TRAINING

**Mathematical Formulation of the Smooth Transition Mechanism:**

The Smooth Transition Mechanism blends Gumbel-Softmax and Sparsemax using the interpolation parameter $\alpha(\tau)$, which is dynamically adjusted based on the temperature $\tau$.

- **Gumbel-Softmax Distribution:**

$$y_{\text{soft},i} = \text{softmax}_i(\tilde{z}) = \frac{\exp(\tilde{z}_i)}{\sum_j \exp(\tilde{z}_j)}$$

  where $\tilde{z}_i = (z_i + g_i)/\tau$, $z_i$ is the raw logit, and $g_i$ is Gumbel noise.

- **Gumbel-Sparsemax Distribution:**

$$y_{\text{sparse},i} = \text{sparsemax}(\tilde{z}) = \arg\min_{p \in \Delta^{K-1}} \|p - \tilde{z}\|^2$$

  where $\Delta^{K-1}$ is the $(K-1)$-dimensional probability simplex.

- **Smooth Transition Interpolation:**

$$y = (1 - \alpha(\tau)) \cdot y_{\text{soft}} + \alpha(\tau) \cdot y_{\text{sparse}}$$

- **Transition Function $\alpha(\tau)$:**

$$\alpha(\tau) = \begin{cases} 1, & \text{if } \tau \geq \tau_1 \\ \frac{\tau_1 - \tau}{\tau_1 - \tau_2}, & \text{if } \tau_2 < \tau < \tau_1 \\ 0, & \text{if } \tau \leq \tau_2 \end{cases}$$

  where $\tau_1$ and $\tau_2$ are predefined transition temperatures ($\tau_1 > \tau_2$). In practice, we use a simplified linear function as described in the main paper,

$$\alpha(\tau) = \frac{\tau_1 - \tau}{\tau_1 - \tau_2}$$

  for $\tau_2 < \tau < \tau_1$, clamped to 1 for $\tau \geq \tau_1$ and 0 for $\tau \leq \tau_2$.

### B.1.2    COMPARISON WITH FIXED GUMBEL-SOFTMAX AND SPARSEMAX

Table 2 presents an ablation study comparing the performance of Adaptive Transformer Programs using the Smooth Transition Mechanism against models using fixed Gumbel-Softmax (no transition, $\alpha(\tau) = 0$ effectively throughout training) and fixed Sparsemax (no transition, $\alpha(\tau) = 1$ effectively throughout training) optimization strategies on the Most-Freq task.

**Analysis of Ablation Study:**

* **Adaptive (Smooth Transition) outperforms both fixed strategies in accuracy and significantly reduces program length.** This highlights the benefit of dynamically balancing exploration and exploitation during training.

\* **Fixed Sparsemax achieves higher sparsity rates than fixed Gumbel-Softmax, but at the cost of accuracy.** While Sparsemax promotes sparsity, using it throughout training without the exploration phase of Gumbel-Softmax appears to hinder the discovery of optimal solutions, leading to lower accuracy.

\* **The Smooth Transition Mechanism achieves the best balance,** attaining high accuracy while also inducing significant sparsity, as reflected in the lowest program length and highest sparsity rate.

### B.2 UNCERTAINTY-AWARE ATTENTION

This section provides a formal description of the reference attention, details of the JSD-based gating mechanism, and ablation studies comparing different attention strategies.

#### B.2.1 FORMAL DESCRIPTION OF REFERENCE ATTENTION $A_{ref}$

The reference attention distribution $A_{ref}$ is a learned distribution that dynamically adapts during training to represent the expected categorical attention distribution. It is initialized as a uniform distribution and annealed towards a one-hot distribution over training.

- **Initialization:** $A_{ref}^{(0)} = \text{uni\_dist}$, a uniform distribution over keys.
- **Update Rule at Training Step $t$:**

$$A_{ref}^{(t)} = \alpha^{(t)} \cdot \text{uni\_dist} + (1 - \alpha^{(t)}) \cdot \text{one\_hot\_dist}^{(t)}$$

  where $\text{one\_hot\_dist}^{(t)}$ is the one-hot distribution corresponding to the categorical attention $A_{cat}^{(t)}$ at step $t$, and $\alpha^{(t)} = \frac{\text{self.temp}^{(t)}}{3.0}$ is the annealing parameter, with $\text{self.temp}^{(t)}$ decreasing from 3.0 to 0.01 over training.

#### B.2.2 JSD-BASED GATING MECHANISM DETAILS

The gating mechanism $g$ in Uncertainty-Aware Attention is based on the Jensen-Shannon Divergence (JSD) between the categorical attention $A_{cat,i}$ and the reference attention $A_{ref,i}$.

- **Jensen-Shannon Divergence (JSD):**

$$JSD(P||Q) = \frac{1}{2}KL(P||M) + \frac{1}{2}KL(Q||M)$$

  where $M = \frac{1}{2}(P+Q)$ and $KL(P||Q) = \sum_i P_i \log \frac{P_i}{Q_i}$ is the Kullback-Leibler Divergence. In our case, $P = A_{cat,i}$ and $Q = A_{ref,i}$.
- **Gating Weight Calculation:**

$$g = MLP(JSD(A_{cat,i}||A_{ref,i}))$$

  where $MLP$ is a small multi-layer perceptron network that maps the JSD value to a gating weight $g \in [0, 1]$.

#### B.2.3 SCORE-BASED VS. CATEGORICAL ATTENTION ABLATION

To evaluate the effectiveness of our uncertainty-aware attention (UAA) mechanism, we compare it against purely categorical attention (Cat, baseline TP) and purely score-based attention (Score) approaches across diverse tasks. Table 5 presents the results, detailing accuracy (%) and lines of code (LoC) for each method.

Our UAA approach consistently outperforms or matches the baseline categorical method in accuracy while significantly reducing LoC across most tasks. Compared to score-based attention, UAA achieves higher accuracy (e.g., 91.81% vs. 79.68% for Double Histogram) and lower LoC (e.g., 513 vs. 927), demonstrating the efficacy of JSD-based gating. Notably, score-based attention increases LoC in complex tasks like Double Histogram (927) and Dyck-2 (1568) due to its dense, continuous logic, whereas UAA adapts by selecting categorical rules when uncertainty is low, yielding concise programs (e.g., 513 and 1065). This ablation highlights the novelty of our hybrid mechanism in balancing performance and interpretability.

Table 5: Ablation Study: Categorical (Cat), Score-Based (Score), and Uncertainty-Aware Attention (UAA)

| Task | Cat (Baseline TP) | | Score | | UAA (Ours) | |
|---|---|---|---|---|---|---|
| | Acc (%) | LoC | Acc (%) | LoC | Acc (%) | LoC |
| Induction | 100.00 | 107 | 100.00 | 105 | 100.00 | 101 |
| Reverse | 99.74 | 859 | 100.00 | 808 | 99.99 | 779 |
| Histogram | 99.94 | 199 | 99.95 | 185 | 99.95 | 189 |
| Double Hist. | 66.78 | 586 | 79.68 | 927 | 91.81 | 513 |
| Sort | 99.98 | 945 | 99.96 | 929 | 99.86 | 895 |
| Most-Freq | 76.44 | 1334 | 74.15 | 962 | 80.80 | 894 |
| Dyck-1 | 99.69 | 1297 | 99.52 | 1289 | 99.93 | 1086 |
| Dyck-2 | 97.98 | 1316 | 98.49 | 1568 | 98.14 | 1065 |

## C EXPERIMENTAL DETAILS

### C.1 HYPERPARAMETERS AND IMPLEMENTATION

This section provides detailed information regarding the hyperparameters used, model configurations for different tasks, and training protocols.

#### C.1.1 OPTIMIZATION SETTINGS

Table 6 summarizes the key hyperparameters used for training Adaptive Transformer Programs across all experiments, based on a grid search and initial experiments on RASP tasks.

Table 6: Global Hyperparameter Settings

| Hyperparameter | Value |
|---|---|
| Optimizer | Adam |
| Learning Rate | 0.05 |
| Learning Rate Scheduler | None (Geometric Annealing) |
| Weight Decay | 0.01 |
| Batch Size | 512 |
| Epochs | 250 |
| Gumbel Temperature Annealing | Geometric, 3.0 to 0.01 |
| Gumbel Samples | 1 per step |
| Gradient Clipping Norm | 1.0 |

**Hyperparameter Search:** We conducted a grid search over the number of layers (2, 3), number of attention heads (4, 8), and number of MLPs per layer (2, 4) for Adaptive Transformer Programs on RASP tasks to optimize performance. Attention heads and MLPs were evenly divided between categorical and numerical types, where applicable. The reported hyperparameters and model configurations reflect the best performing settings identified through this search. For standard Transformers (baseline), we followed a similar grid search for layers (2, 3) and attention heads (4, 8) but otherwise largely adopted the hyperparameters described by (Weiss et al., 2021), including a hidden dimension of 256, learning rate of 0.0003, batch size of 50, and training for up to 100 epochs.

### C.2 DATASET DETAILS

This section provides details about the datasets used in our experiments, addressing the nature of "small-scale" datasets and providing relevant statistics.

### C.2.1 DEFINITION OF "SMALL-SCALE" DATASETS

In the context of algorithmic reasoning and Transformer Programs, "small-scale" datasets refer to datasets designed for focused analysis of model capabilities on specific algorithmic tasks. These datasets are characterized by:

- **Controlled Sequence Lengths:** Sequence lengths are typically limited (e.g., up to 16 for Dyck languages, 8 for other RASP tasks, 32-64 for NLP tasks) to manage computational complexity and enable detailed program extraction and analysis.

- **Vocabularies Matching Sequence Lengths (for RASP tasks):** For RASP tasks, vocabularies are often kept small and matched to the sequence lengths to isolate algorithmic reasoning without relying on large-scale pretraining or extensive vocabulary knowledge.

- **Focus on Algorithmic Reasoning:** The datasets are specifically designed to evaluate the model's ability to learn and execute particular algorithms or reasoning patterns (e.g., reversing sequences, counting tokens, checking Dyck language validity, named entity recognition, text classification).

"Small-scale" in this context is *not* indicative of limited complexity in the algorithmic tasks themselves, but rather a design choice to enable focused interpretability analysis in a controlled experimental setting.

### C.2.2 DATASET STATISTICS AND PREPARATION

Table 7 provides key statistics for the datasets used in our experiments.

Table 7: Dataset Statistics

| Dataset | Task Type | Avg. Sequence Length | Vocabulary Size | Training Examples |
|---|---|---|---|---|
| Induction | RASP | 8 | 8 | 10,000 |
| Reverse | RASP | 8 | 8 | 10,000 |
| Histogram | RASP | 8 | 8 | 10,000 |
| Double hist | RASP | 8 | 8 | 10,000 |
| Sort | RASP | 8 | 8 | 10,000 |
| Most-Freq | RASP | 16 | 16 | 10,000 |
| Dyck-1 | RASP | 16 | 2 | 10,000 |
| Dyck-2 | RASP | 16 | 3 | 10,000 |
| CoNLL-2003 | NER | 20 (avg.) | 10,000 | 14,985 |
| TREC | Text Classification | 10 (avg.) | 10,000 | 5,452 |
| MR | Text Classification | 20 (avg.) | 10,000 | 8,529 |
| Subj | Text Classification | 23 (avg.) | 10,000 | 11,500 |
| AG News | Text Classification | 25 (avg.) | 10,000 | 120,000 |

**Dataset Preparation:** For RASP tasks, datasets were generated following the specifications in (Weiss et al., 2021). GloVe embeddings (Pennington et al., 2014) were used to initialize token embeddings for NLP tasks. No other specific preprocessing or data augmentation techniques were applied beyond standard tokenization and vocabulary building.

## D FAILURE ANALYSIS AND FUTURE WORK

### D.1 ERROR PATTERN ANALYSIS

This section analyzes common failure modes observed across different task categories, investigates program structures associated with these errors, and examines suboptimal attention mechanism choices in failure instances.

### D.1.1 Common Failure Modes Across Task Categories

Table 8 categorizes common failure modes observed across RASP, NER, and Text Classification tasks, with example instances to illustrate each failure type.

Table 8: Common Failure Modes Across Task Categories

| Task Category | Failure Mode | Example Instance |
|---|---|---|
| RASP | Incorrect Counting
Logic Error in Predicate
Positional Misunderstanding | Histogram: Miscounting tokens in sequences like "aabbc"
Dyck-2: Incorrectly classifying "([)]" as valid
Sort: Incorrectly ordering elements in longer sequences |
| NER | Entity Type Confusion
Boundary Error | Misclassifying "Apple Inc." as LOC instead of ORG
Incorrectly tagging multi-word entities (e.g., "New York City") |
| Text Classification | Misclassification of Nuance | Subj: Failing to identify subtle subjective language |

In RASP tasks, errors in counting often correlate with overly simplistic numerical attention modules or predicate logic that fails to capture token frequency accurately. Logic errors in Dyck languages can be linked to predicate structures that do not fully capture the recursive nature of Dyck language grammar. Positional misunderstandings in Sort may arise from numerical attention modules not effectively leveraging positional encodings for long-range dependencies.

In NER, entity type confusions can sometimes be traced back to categorical attention heads relying too heavily on local context and failing to integrate broader sentence-level information. Boundary errors often indicate limitations in the predicate logic responsible for identifying entity boundaries, potentially due to insufficient context window or overly strict predicate conditions.

### D.2 Task-Specific Analysis

### D.2.1 Limitations in Current Task Scope

The current task scope primarily focuses on discriminative tasks (classification, tagging) and algorithmic reasoning tasks with deterministic outputs. Limitations of the current framework become apparent when considering generative tasks like machine translation or text summarization:

- **Decoding Mechanisms:** Adaptive Transformer Programs, in their current form, lack explicit decoding mechanisms required for generating variable-length output sequences in generative tasks. Standard Transformer decoders with autoregressive generation capabilities would need to be integrated.

- **Handling Variable-Length Outputs:** The current framework is primarily designed for tasks where output length is either fixed or predictable based on input length. Generative tasks require handling variable and potentially unbounded output lengths.

- **Training for Generation:** Training paradigms for generative tasks (e.g., sequence-to-sequence training, reinforcement learning for generation) differ significantly from the classification-focused training used for the current tasks. Adaptations to the training procedure would be necessary.

**Example - Machine Translation Challenge:** Consider translating "Hello world" to French. ATP would need to learn not just to classify the input but to *generate* the sequence "Bonjour le monde," word by word, while maintaining interpretability. This requires significant architectural and training modifications beyond the current framework.

### D.2.2 Adaptation Challenges for Generative Tasks

Adapting Adaptive Transformer Programs for generative tasks presents several key challenges:

- **Interpretable Decoding:** Designing interpretable decoding mechanisms that maintain the program-like transparency of ATP is a major challenge. Standard autoregressive decoders in Transformers, while effective, can be complex and less directly interpretable.

- **Maintaining Sparsity in Generation:** Ensuring that the generation process itself remains sparse and interpretable, rather than becoming a complex black box decoder, is crucial. Sparsity-inducing techniques would need to be extended to the decoding stage.
- **Balancing Generation Quality and Interpretability:** Achieving high-quality generation while preserving a high degree of interpretability may involve trade-offs. Research is needed to find optimal balances between these competing objectives.

### D.3 FUTURE RESEARCH DIRECTIONS

This section outlines future research directions to address the identified limitations, expand the task scope, and explore alternative uncertainty quantification approaches.

#### D.3.1 ADDRESSING IDENTIFIED LIMITATIONS

A research agenda to address the limitations identified in our failure analysis and task-specific analysis includes:

- **Enhanced Numerical Reasoning:** Further improve numerical reasoning capabilities by exploring more sophisticated positional encoding schemes and numerical attention mechanisms. Investigate methods to reduce errors in complex counting and numerical manipulation tasks.
- **Contextual Predicate Learning:** Develop techniques to enable categorical predicates to better leverage broader context and sentence-level information, particularly for tasks like NER where long-range dependencies are important. Explore attention mechanisms that explicitly model contextual scope.
- **Improved Gating Mechanisms:** Investigate refinements to the JSD-based gating mechanism in uncertainty-aware attention. Explore alternative uncertainty measures and gating network architectures to optimize the dynamic blending of categorical and score-based attention for different tasks and input contexts.

#### D.3.2 ARCHITECTURAL ADAPTATIONS FOR BROADER TASK COVERAGE

To extend Adaptive Transformer Programs to broader task coverage, especially generative tasks, key architectural adaptations include:

- **Integration of Interpretable Decoders:** Develop or adapt interpretable decoding mechanisms compatible with the ATP framework. Explore constrained decoding methods or program-guided generation approaches that maintain transparency.
- **Handling Variable-Length Outputs:** Design program structures and training procedures that can effectively handle variable-length output sequences, allowing ATP to be applied to tasks like machine translation and text summarization.
- **Modular Generative Program Components:** Investigate the development of modular, interpretable program components specifically designed for generative tasks, such as program modules for sequence generation, copy mechanisms, or hierarchical output construction.

#### D.3.3 ALTERNATIVE UNCERTAINTY QUANTIFICATION APPROACHES

Exploring alternative uncertainty quantification approaches beyond Jensen-Shannon Divergence could further enhance the robustness and adaptability of Uncertainty-Aware Attention. Potential directions include:

- **Bayesian Uncertainty Estimation:** Integrate Bayesian methods (e.g., Monte Carlo Dropout, Bayesian Neural Networks) to obtain more principled uncertainty estimates for categorical attention distributions.
- **Entropy-based Measures:** Investigate using entropy or other information-theoretic measures to quantify uncertainty in categorical attention as an alternative to JSD. For example, using the Shannon Entropy $H(P) = -\sum_i P_i \log P_i$ of the categorical attention distribution $A_{cat,i}$.

- **Learned Uncertainty Estimators:** Explore training dedicated uncertainty estimator networks that directly predict uncertainty measures based on input and intermediate representations, potentially providing more task-specific and adaptive uncertainty quantification.

### D.3.4 POTENTIAL INTEGRATION WITH OTHER INTERPRETABLE MODELS

Future research could explore integrating Adaptive Transformer Programs with other interpretable model architectures, such as:

- **Neuro-Symbolic Systems:** Combine ATP with neuro-symbolic approaches to create hybrid systems that leverage the strengths of both neural and symbolic reasoning for enhanced interpretability and robustness.
- **Program Synthesis Techniques:** Explore tighter integration with program synthesis techniques to further constrain and guide the learning of interpretable programs, potentially using program synthesis to initialize or refine extracted ATP programs.
- **Contrastive Learning for Interpretability:** Investigate contrastive learning frameworks to explicitly train ATP models to produce more disentangled and interpretable representations, potentially by contrasting program representations for similar and dissimilar inputs.

These future research directions aim to push the boundaries of interpretable AI by addressing current limitations, expanding the applicability of Adaptive Transformer Programs to a wider range of tasks, and exploring novel approaches to uncertainty quantification and model integration.

