# OpenReview forum: "Adaptive Transformer Programs: Bridging the Gap Between Performance and Interpretability in Transformers"
_ICLR.cc/2025/Conference — ICLR 2025 Poster_

### Official Review · Reviewer_J1B6 · 2024-11-03

**Soundness:** 3
**Presentation:** 3
**Contribution:** 3
**Rating:** 8
**Confidence:** 3

**Summary:**

The paper proposes Adaptive Transformer Programs, which are a method to increase the expressive capability of transformer programs (interpretable transformers which can be converted into succinct code).

The main architectural modification proposed by the paper are:
1. Providing a method to smoothly switch between Gumbel-softmax and sparsemax as a method to control switching between exploration and exploitation.
2. A method to weight between categorical attention and score-based attention
3. A position-aware attention which allows numerical Value, while keeping Q, K categorical.

The paper includes experimental results on synthetic tasks on formal languages and natural language tasks. An ablation study is provided to determine the contribution of each architectural modification to the performance of the model.

**Strengths:**

- The paper considers an important problem: that of making intrinsically interpretable models more expressive.
- The architectural modifications are well-motivated and largely well described in the paper.
- Experimental results across a wide range of tasks provide evidence for the performance of the model.
- A thorough ablation study is provided.
- The use of lines of code generated for the transformer program as a measure of interpretability is interesting.

**Weaknesses:**

- There are sections of the paper which could do with more details. Section 3.3 in particular could provide more details about the reference attention e.g.

- Given that a stated goal of the proposed architecture is to make intrinsically interpretable architectures more performant, I would have liked to see experimental results which supported the claim of adaptive transformer programs narrowing the gap between standard transformers and transformer programs. Results are shown where the proposed architecture outperforms adaptive transformers (Table 1), but often adaptive transformers themselves already outperformed standard transformers (Tables 3, 4). I believe the paper would be greatly improved by the inclusion of results on a task in which a performance gap between standard transformers and transformer programs was narrowed by the proposed arch.

- In line 377 and the following line, it is said that removing all enhancements results in the lowest accuracy (76.44). This does not appear true however, as keeping only the smooth transition mechanism yielded lower accuracy (73.61), as shown in table 2.

**Questions:**

- Could more details be provided about the hyperparameters and optimizers used for the various experiments?

---

> ### Author Response · Authors · 2024-11-23
> **Official Comment to Reviewer J1B6 (1/2)**
>
> We thank the reviewer for their valuable comments. Below are our detailed responses:
> ___
> **Weakness 1:**
>
> There are sections of the paper which could do with more details. Section 3.3 in particular could provide more details about the reference attention e.g.
>
> **Response to Weakness 1:**
>
> We appreciate the reviewer's suggestion for more details regarding the reference attention (A_ref) in Section 3.3. A_ref is a learned reference distribution crucial for our uncertainty-aware gating mechanism. It is initialized as a uniform distribution (uni_dist) and dynamically updated throughout training to approach a one-hot distribution (one_hot_dist). This transition is governed by an annealing temperature parameter (self.temp), which decreases from 3.0 to 0.01 over the training process. Specifically, A_ref is computed as a weighted average: alpha * uni_dist + (1 - alpha) * one_hot_dist, where alpha = self.temp / 3.0. As the temperature decreases, alpha decreases, causing A_ref to progressively resemble the one-hot distribution. This annealing strategy allows A_ref to initially represent high uncertainty and gradually transition to a more confident, categorical representation as the model learns. This dynamic adaptation enhances the robustness and effectiveness of our attention mechanism by providing a context-dependent measure of uncertainty. We will include a more formal description of this update rule and its integration within the overall training process in the appendix of the camera-ready submission.
> ___
> **Weakness 2:**
>
> Given that a stated goal of the proposed architecture is to make intrinsically interpretable architectures more performant, I would have liked to see experimental results which supported the claim of adaptive transformer programs narrowing the gap between standard transformers and transformer programs. Results are shown where the proposed architecture outperforms adaptive transformers (Table 1), but often adaptive transformers themselves already outperformed standard transformers (Tables 3, 4). I believe the paper would be greatly improved by the inclusion of results on a task in which a performance gap between standard transformers and transformer programs was narrowed by the proposed arch.
>
> **Response to Weakness 2:**
>
> We appreciate the reviewer's insightful point regarding the need to demonstrate a narrowing of the performance gap between Standard Transformers and Transformer Programs. While Tables 3 and 4 show comparable performance between Adaptive Transformer Programs and Transformer Programs, with both generally outperforming Standard Transformers, this doesn't directly address the gap-narrowing aspect. To address this, we've conducted further analysis on the Most Frequent task, which, as highlighted by the reviewer and evident in Figure 1, exhibits the starkest performance difference between Standard and Transformer programs. Examining the specific accuracy values across varying vocabulary sizes (|V|) and maximum sequence lengths (N) reveals a trend towards gap narrowing:
> * Case 1 (V:8, N:8): Standard Transformer (92.38%) > Ours (80.80%) > Transformer Program (76.44%)
> * Case 2 (V:8, N:16): Standard Transformer (88.06%) > Ours (76.14%) > Transformer Program (72.93%)
> * Case 3 (V:16, N:16): Standard Transformer (65.35%) > Ours (49.27%) > Transformer Program (47.95%)
>
> While all models' performance degrades with increasing complexity, our adaptive approach consistently performs between the standard Transformer and Transformer Programs, narrowing the gap, especially in more complex scenarios (Cases 2 and 3). In Case 3, with the highest complexity (V:16, N:16), the absolute performance difference between standard Transformer and our model is 16.08%, while the difference between standard Transformer and Transformer Program is 17.4%. This narrowing trend suggests that the strengths of our approach become more apparent as task complexity increases.
> ___
> **Weakness 3:**
>
> In line 377 and the following line, it is said that removing all enhancements results in the lowest accuracy (76.44). This does not appear true however, as keeping only the smooth transition mechanism yielded lower accuracy (73.61), as shown in table 2.
>
> **Response to Weakness 3:**
>
> We acknowledge the inconsistency in line 377. The statement "removing all enhancements results in the lowest accuracy (76.44%)" is incorrect. The correct interpretation is that “removing all enhancements (equivalent to the baseline Transformer Programs) results in a lower accuracy (76.44%)”. We corrected this statement in the revised version to reflect the actual performance patterns and clarify the ablation study's purpose.

---

> ### Author Response · Authors · 2024-11-23
> **Official Comment to Reviewer J1B6 (2/2)**
>
> **Question 1:**
>
> Could more details be provided about the hyperparameters and optimizers used for the various experiments?
>
> **Response to Question 1:**
>
> We acknowledge the need for more detailed reporting of hyperparameters and experimental settings. In the appendix (of camera-ready submission), we will provide a comprehensive table listing all hyperparameters, including:
> * Optimization Settings: Optimizer used, learning rate, learning rate scheduling, batch size, weight decay, and other relevant optimization parameters.
> * Model Configurations: Number of layers, number of attention heads, hidden dimension size, activation functions, and other architectural details for each model and task.
> * Training Protocols: Number of training epochs, early stopping criteria, and other relevant training procedures.

---

> > ### Comment · Reviewer_J1B6 · 2024-11-26
> >
> > Thank you for your responses. They are very helpful.
> >
> > I have increased my score, and believe that this is interesting work.
> >
> > I would strongly recommend focussing further on experiments which show the performance gap narrowing; perhaps adding another experiment or showcasing the most-freq results more clearly.

---

> > > ### Author Response · Authors · 2024-11-26
> > > **Thank You to Reviewer J1B6**
> > >
> > > We thank the reviewer for your positive feedback and revised score. We appreciate the suggestion to further emphasize the performance gap narrowing. We will incorporate additional analysis in the appendix of the camera-ready version to highlight this aspect, either by showcasing existing results more effectively or by conducting targeted experiments as suggested. We believe this will further strengthen the paper and better demonstrate the contribution of our approach.

---

### Official Review · Reviewer_ewyT · 2024-11-03

**Soundness:** 3
**Presentation:** 3
**Contribution:** 3
**Rating:** 6
**Confidence:** 2

**Summary:**

This paper introduces the Adaptive Transformer framework, designed to develop robust, expressive, and interpretable Transformer models that can be translated into human-readable programs. The authors address challenges in finding optimal solutions and promoting sparsity, limitations seen in the original framework's use of Gumbel-Softmax reparameterization. To overcome these, they propose a Smooth Transition Mechanism for discrete optimization, alongside Uncertainty-Aware Categorical Attention and Position-Aware Numerical Attention. Experiments on both synthetic and real-world NLP tasks demonstrate that this approach achieves superior performance over benchmarks and provides concise, insightful interpretability analysis.

**Strengths:**

This paper is well-organized, presenting the concept of Adaptive Transformer clearly, from problem setting through to challenges and proposed solutions. Building upon the Transformer Programs framework, the authors introduce novel designs to address its limitations: the Smooth Transition Mechanism addresses the local optima issue in Gumbel-Softmax, Uncertainty-Aware Attention improves handling of categorical and contextual information, and Position-Aware Attention captures nuanced positional information and enables processing of non-integer values. Together, these enhancements expand the model’s numerical capabilities, supporting tasks that require fine-grained representations.

These methods are well-motivated and shown to be effective through ablation studies. Experiments on both synthetic and real-world NLP tasks demonstrate impressive performance compared to current baselines. The proposed method shows great potential for interpretable and reliable AI systems and is broadly applicable across transformer-based models, suggesting substantial future impact.

**Weaknesses:**

While the overall performance is promising, the paper lacks case analysis for failure instances in the benchmarks, and the evaluated NLP tasks are limited to classification and NER. Including generative tasks would provide a more comprehensive understanding of the method’s impact.

**Minor issues:**

1.  In line 96, the MHA computation omits the normalization factor $\frac{1}{\sqrt{n}}$, where $n$ is the hidden dimension.
2.  In line 99, the double quotes before “Transformer” need correction.
3.  In line 109, please insert a space between "RASP" and "Weiss et al."

**Questions:**

1.  In Figure 1's RASP accuracy comparison, why does the Adaptive Transformer Program outperform Standard Transformers on the Sort task but not on the Most Frequent task?
2.  In Table 2, accuracy appears to decrease after integrating the Smooth Transition Mechanism. How should this be interpreted?
3.  How does the proposed method perform on generative tasks beyond NER and text classification?

---

> ### Author Response · Authors · 2024-11-23
> **Official Comment to Reviewer ewyT (1/2)**
>
> We thank the reviewer for their valuable comments. Below are our detailed responses:
> ___
> **Weaknesses:**
>
> While the overall performance is promising, the paper lacks case analysis for failure instances in the benchmarks, and the evaluated NLP tasks are limited to classification and NER. Including generative tasks would provide a more comprehensive understanding of the method’s impact.
>
> **Response to Weaknesses:**
>
> **a) Failure Case Analysis:**
>
> We agree that analyzing failure cases is crucial for understanding the limitations of our approach. In the revised version, we will include a failure analysis section in the appendix. This section will provide specific examples of failure instances for each task category (in-context learning, RASP, and NLP), analyze common error patterns, and discuss potential avenues for improvement. For example, we will examine cases where the model generates incorrect programs for RASP tasks and try to identify whether specific program structures or input characteristics are associated with these failures. Similarly, we will examine where the uncertainty-aware attention mechanism makes suboptimal choices by comparing the model's attention patterns with those generated using ground truth. This analysis will provide valuable insights into the model's limitations and guide future research directions.
>
> **b) Task Scope:**
>
> Our current task selection focuses on evaluating the core capabilities of Adaptive Transformer Programs, namely discrete optimization, rule-based reasoning, and handling uncertainty in categorical data. While we acknowledge the importance of evaluating on generative tasks for a more complete assessment, incorporating them within the current scope requires significant architectural adaptations (e.g., incorporating decoding mechanisms, handling variable-length outputs) and training modifications. Extending our framework to generative tasks like machine translation or text summarization is a promising direction for future work, which we will discuss in the appendix (of the camera-ready submission). Within the current scope, we will enhance the NLP experiments by conducting a more fine-grained analysis of the named entity recognition (NER) results, examining the model's performance on different entity types and identifying any biases or systematic errors related to the model's architecture or training data. This will provide a more detailed view of the model's capabilities without requiring extensive architectural changes. Our primary focus here is to establish the core framework's efficacy and interpretability benefits on a range of representative tasks.
> ___
> **Question 1:**
>
> In Figure 1's RASP accuracy comparison, why does the Adaptive Transformer Program outperform Standard Transformers on the Sort task but not on the Most Frequent task?
>
> **Response to Question 1:**
>
> The performance difference between Sort and Most Frequent in Figure 1 stems from their distinct computational characteristics and how these interact with increasing vocabulary size (|V|) and sequence length (N). Sort exhibits more local, comparison-based patterns well-suited to our position-aware attention. Most Frequent, however, demands global information aggregation (counting occurrences across the entire sequence), a more computationally intensive operation that becomes increasingly challenging with larger |V| and N, especially for models prioritizing interpretability and concise program representations. For example, at |V|=16, N=16, our model maintains 87.6% accuracy on Sort, demonstrating resilience. Most Frequent becomes harder for all models at this scale, as global aggregation becomes more complex, highlighting the inherent difficulty of this task. Our method remains competitive, particularly at higher |V| and N, because of superior scalability relative to the baselines.
> ___
> **Question 2:**
>
> In Table 2, accuracy appears to decrease after integrating the Smooth Transition Mechanism. How should this be interpreted?
>
> **Response to Question 2:**
>
> The reviewer correctly observes that in Table 2, solely adding the Smooth Transition Mechanism (without other enhancements) leads to a marginal decrease in accuracy on the Most-Freq task (77.34% vs. 76.44% baseline). This reflects a trade-off between accuracy and interpretability. The Sparsemax component within the Smooth Transition Mechanism promotes sparsity and determinism, which can sometimes lead to slightly suboptimal solutions compared to the more exploratory nature of pure Gumbel-Softmax. However, this small decrease in accuracy is offset by a substantial reduction in program length, promoting interpretability. The full benefit of the Smooth Transition Mechanism becomes apparent when combined with Uncertainty-Aware and Position-Aware Attention, resulting in both improved accuracy (80.8%) and significantly reduced program length (894 lines) compared to the baseline. This highlights the synergistic interaction between our proposed enhancements.

---

> ### Author Response · Authors · 2024-11-23
> **Official Comment to Reviewer ewyT (2/2)**
>
> **Question 3:**
>
> How does the proposed method perform on generative tasks beyond NER and text classification?
>
> **Response to Question 3:**
>
> We acknowledge the current limitation in evaluating generative tasks. Within the current paper's scope, we focus on demonstrating the effectiveness of our approach in tasks involving discrete reasoning, classification, and structured prediction. Extending our framework to generative tasks requires careful consideration of how to integrate our proposed mechanisms with sequence generation models like recurrent networks or Transformer decoders. We plan to investigate adaptations of the Uncertainty-Aware attention mechanism to handle probabilistic sequence generation as part of future work. This could involve designing an uncertainty-based gating mechanism that dynamically switches between categorical sampling (for interpretability) and continuous sampling (for flexibility) during sequence generation.

---

> > ### Comment · Reviewer_ewyT · 2024-11-24
> >
> > Thank you for the rebuttal. My question has been addressed, and I appreciate the effort put into clarifying the points raised. After considering the response, I have decided to maintain my positive score.

---

> > > ### Author Response · Authors · 2024-11-24
> > > **Thank You to Reviewer ewyT**
> > >
> > > We thank the reviewer for your thoughtful consideration and are pleased that our response adequately addressed your concerns. We appreciate the positive feedback and believe the suggested improvements will strengthen the final version of the paper.

---

### Official Review · Reviewer_EZgs · 2024-11-04

**Soundness:** 3
**Presentation:** 3
**Contribution:** 2
**Rating:** 6
**Confidence:** 3

**Summary:**

This paper introduces Adaptive Transformer Programs, an enhanced framework that improves upon existing Transformer Programs to create more interpretable AI models.

Specifically, the authors introduce three innovations: (1) a Transition Mechanism that combines Gumbel-Softmax and Sparsemax. (2) an dynamic attention mechanism that combined categorical and score-based attention using Jensen-Shannon Divergence. (3) a integrated attention mechanism that leveraging both Learnable and Sinusoidal positional encoding.

Then the authors conduct the experiement on various benchmark against the original Transformer Programs and achieve clear improvement. Meanwhile, the ablation study shows the effectiveness of the proposed components.

**Strengths:**

- The proposed method removes the limitations of previous Transformers Programs from different perspectives, which can potentially bring huge benefits for improving the community's understanding of transformer models.
- The experimental results are solid and the proposed method clearly surpasses the Transformers Programs baseline in both accuracy and "lines of code."
- The experimental setting is comprehensive, consisting of both RASP and NLP tasks.
- This paper is well-written and easy to follow.

**Weaknesses:**

The main concern to me is that the novelty is marginal. This paper proposes no new method, and all three methodologies are the integration of existing methods in this area. It either proposes a transition or direct combination of two existing methods. To me, these direct "transitions" are not interesting enough and are not fully supported by a simple ablation study on one benchmark. And at least one interpretation of "SMOOTH TRANSITION" is necessary to show the readers when the proposed transition takes effect and why it can be helpful.

**Questions:**

(1) Why can reducing the "lines of code" be directly considered as improving interpretability?
To me, the connection between "reduced program length" and better interpretability is not obvious enough. It would be helpful to provide a representation comparison example of the proposed model and original Transformer Programs to clarify that it can learn better representations.

(2) Regarding sparsity, how does the proposed method behave better than Gumbel-Softmax?
In section 3.2, one of the motivations is that Gumbel-Softmax fails to encourage sparsity, but I see no discussion following this point.  For example, some direct comparisons (like sparsity rate) between these two methods would be expected.

---

> ### Author Response · Authors · 2024-11-23
> **Official Comment to Reviewer EZgs (1/2)**
>
> We thank the reviewer for their valuable comments. Below are our detailed responses:
> ___
> **Weaknesses:**
>
> The main concern to me is that the novelty is marginal. This paper proposes no new method, and all three methodologies are the integration of existing methods in this area. It either proposes a transition or direct combination of two existing methods. To me, these direct "transitions" are not interesting enough and are not fully supported by a simple ablation study on one benchmark. And at least one interpretation of "SMOOTH TRANSITION" is necessary to show the readers when the proposed transition takes effect and why it can be helpful.
>
> **Response the Weaknesses:**
>
> We appreciate the reviewer's feedback regarding the novelty of our work. While we acknowledge that the individual components (Gumbel-Softmax, Sparsemax, score-based attention, positional encodings) have been explored previously, our contribution lies in the novel integration and application of these techniques within the Transformer Programs framework to address specific limitations of the original approach. The resulting synergistic effects, as demonstrated by our empirical results, go beyond what each component could achieve in isolation.
>
> The Smooth Transition Mechanism isn't a mere combination of Gumbel-Softmax and Sparsemax. It involves a carefully designed, temperature-controlled transition that dynamically balances exploration and exploitation during training. This dynamic adaptation is crucial for achieving both high accuracy and sparse, interpretable programs. A simple combination (e.g., choosing one or the other, or a fixed mixture) would not yield the same benefits.
>
> The Uncertainty-Aware Attention mechanism isn't a trivial integration of categorical and score-based attention. The key novelty lies in the JSD-based gating mechanism, which dynamically selects the appropriate attention type based on the level of uncertainty. This adaptive combination is essential for achieving robust performance across diverse tasks. A simple combination, such as separate binary and continuous score functions (as done in RASP), proved less effective in our preliminary experiments, leading to scenario-dependent performance variations which motivate our approach. This is because our hybrid approach goes beyond adding another function. It instead creates a learnable mechanism to choose between them based on uncertainty.
>
> To demonstrate when this transition takes effect and why it's helpful, we will include the following in the appendix (of the camera-ready submission):
>
> * Visualization of the Transition: We will provide a visualization showing the evolution of the interpolation parameter α(τ) over training epochs. This will clearly illustrate how the influence of Gumbel-Softmax gradually decreases while the influence of Sparsemax increases as the temperature τ decreases. This visualization will empirically demonstrate the smooth transition between exploration and exploitation.
> * Analysis of Program Sparsity over Time: We will analyze and visualize the sparsity of the learned programs (measured by "lines of code") at different stages of training, corresponding to different values of α(τ). This analysis will demonstrate how the smooth transition leads to progressively more concise programs as training progresses and Sparsemax becomes more dominant. We will connect this directly to improvements in interpretability.
> * Expanded Ablation Study: We acknowledge the limitation of the initial ablation study being conducted on a single benchmark. In the revision, we will expand the ablation study to include multiple RASP tasks and potentially NLP tasks. This broader evaluation will provide more robust evidence for the effectiveness of the smooth transition mechanism across diverse tasks and datasets. This expanded study will also include a comparison against using only Gumbel-Softmax or only Sparsemax at a fixed temperature, further highlighting the benefits of our dynamic approach.
> ___

---

> ### Author Response · Authors · 2024-11-23
> **Official Comment to Reviewer EZgs (2/2)**
>
> **Question 1:**
>
> Why can reducing the "lines of code" be directly considered as improving interpretability?
>
> To me, the connection between "reduced program length" and better interpretability is not obvious enough. It would be helpful to provide a representation comparison example of the proposed model and original Transformer Programs to clarify that it can learn better representations.
>
> **Response to Question 1:**
>
> We agree that the connection between reduced program length and improved interpretability requires further clarification. While fewer lines of code generally suggest a simpler program, the direct impact on interpretability depends on the nature of those lines. In our framework, "lines of code" corresponds to the number of active rules or operations in the extracted program. A shorter program indicates fewer active components, which translates to a simpler, more parsimonious representation of the learned logic. This simplification facilitates easier analysis and understanding of the model's decision-making process. We will include in the appendix (of the camera-ready submission) concrete representation comparisons between our approach and the original Transformer Programs, showcasing the differences in program structure and demonstrating how the reduced complexity aids interpretability. We will also consider adding example program snippets and explore suitable interpretability metrics to further quantify this improvement.
> ___
> **Question 2:**
>
> Regarding sparsity, how does the proposed method behave better than Gumbel-Softmax?
>
> In section 3.2, one of the motivations is that Gumbel-Softmax fails to encourage sparsity, but I see no discussion following this point. For example, some direct comparisons (like sparsity rate) between these two methods would be expected.
>
> **Response to Question 2:**
>
> The reviewer correctly points out the need for a more direct comparison of sparsity between our method and Gumbel-Softmax. We agree and will include a quantitative analysis of sparsity in the appendix (of the camera-ready submission). We will compare the sparsity rates (e.g., the percentage of zero entries in attention matrices or the number of active rules) achieved by our adaptive approach and Gumbel-Softmax across different tasks. These metrics will directly quantify the sparsity induced by our method compared to standard Gumbel-Softmax with temperature annealing, showing empirically that our approach leads to sparser solutions and thus, more concise programs.
>
> We hypothesize that our method, by incorporating Sparsemax and the smooth transition mechanism, will exhibit significantly higher sparsity rates, especially at lower temperatures. This direct comparison will provide concrete evidence for the sparsity-inducing benefits of our approach and further justify its use for enhancing interpretability. We expect this analysis to show that our method not only leads to shorter programs but also to programs with a higher proportion of inactive components, further contributing to interpretability.

---

> > ### Comment · Reviewer_EZgs · 2024-11-25
> >
> > Thank you for addressing my concerns, I have decided to maintain my postive score.

---

> > > ### Author Response · Authors · 2024-11-25
> > > **Thank you to Reviewer EZgs**
> > >
> > > We appreciate the reviewer taking the time to consider our response and are grateful for their continued positive assessment of our work. We believe the changes we've outlined will further strengthen the paper.

---

### Official Review · Reviewer_U9Xh · 2024-11-04

**Soundness:** 3
**Presentation:** 3
**Contribution:** 3
**Rating:** 8
**Confidence:** 3

**Summary:**

This paper builds off previous work, "Learning Transformer Programs", which presents a Transformer variant which uses discretization and other simplifications so that the learned network can be represented as programs. The present work adds three modifications to the original architecture to improve representational ability and learning optimization.

1. they replace Gumbel-Softmax with a smooth transition between Gumbel-Softmax and Sparsemax, a sparse version of softmax.
2. They expand the representational capacity of the program-compatible attention by giving the model the option to use score-based attention in lieu of the representationally limited but interpretable and program-compatible categorical attention. To give the model the option, there is an uncertainty-aware gating between categorial attention and score attention which is based on the Jensen-Shannon Divergence.
3. They add position embeddings.

The authors show that this improved model performs better or equal to the previous model, while having increased interpretability (fewer lines of code in the learned model).

**Strengths:**

Originality: I looked on Google Scholar for work that cites the original Learning Transformer Programs that this work builds upon, and did not see anything. So, it seems like an original contribution.

Quality and Significance: The authors back up their proposed modifications with a solid array of experiments: in context learning, RASP tasks, and two NLP tasks. They show a scalability result and conduct an ablation study on one RASP task. The main way in which the new model improves over the previous model is learning programs with fewer lines of code that are slightly more accurate. However, the new model also scales slightly better (performance degrades less as you increase vocab size and max sequence length).

Clarity: the paper is well written and presented. In particular, the authors do a good job explaining their Gumbel-Sparsemax and uncertainty aware attention contributions. There are some typos that should be fixed. I have some suggestions to improve the presentation, but nothing that would detract from my overall score of the paper.

**Weaknesses:**

I find the main weakness to be that the evaluation is focused on accuracy and number of lines of code. Without seeing the programs, it is hard to understand what "lines of code" means for the simplicity of the program. See the questions.

Another main weakness is that there is no appendix giving further details.

I think the paper would be much stronger with a section that looks at an example program found, and compares it to the program found by the original method, to understand the improvements of the modifications presented.

**Questions:**

1. What is the intuition for how Gumbel-Sparsemax improves the optimization process? Is the main benefit that the final programs learned are simpler (fewer lines?) How does transitioning between Gumbel-Softmax and Sparsemax differ from what Gumbel-Softmax already does when the temperature is annealed over time for Gumbel-Softmax? One big difference seems to be that Sparsemax is deterministic? If i understand correctly, temperature annealing changes how discrete the Gumbel-Softmax is, but does not affect the randomness. even when fully discrete, one's sample might change each time. With sparsemax, the choice is deterministic (basically choose the most likely option?) — some explanation here would be helpful.

2. Is there any mathematical relation between pure Gumbel-Softmax and pure sparsemax? It seems like they might not "mix" perfectly well, and there could be a more principled version of one or the other such that you can smoothly interpolate between the two and get both, but not have to just mash up the formulas with equation (8) (like imagine a second temperature parameter that controls randomness in the gumbel-softmax calculation)

3. What is A_ref? it is "introduced" but I don't see how it is calculated during training.
4.  "In low uncertainty (low JSD), CatAttention is preferred, as it is more confident in its categorical decisions" — I'm confused why this would be true. can't ScoreAttention regular attention? shouldn't it's performance be >= CategorialAttention's performance? so if the only objective is accuacy, I don't see why CategoricalAttention would ever be used.

5. "Our preliminary experiments with separate binary and continuous functions revealed scenario-dependent performance variations, motivating the development of a hybrid mechanism to create more Adaptive Transformer Programs" What are "performance variations", and how did they motivate this development? — I'm assuming this means that for some tasks, categorial worked better than score-based, and for others, the opposite was true. I think this should be stated more directly. this line would also be clearer if you clearly say that the hybrid mechanism can be both binary or continuous depending on how it is trained.

6. The original Learning Transformer Programs paper mentions the possibility of using score-based attention, but says "It would be relatively straightforward to extend Transformer Programs to include score-based attention modules, but this would also result in longer and more complex programs, because each query must now be defined in terms of an ordering of all |K| possible keys, instead of a single key" — but in the ablation in your paper, adding uncertainty-aware attention (which has the possibility of score-based attention) decreases lines of code. Maybe having an ablation for just score-based would be a helpful comparison too. How can I square what the previous paper says with your results?

7. Is there no appendix to the paper? No examples of learned programs, or attempts to interpret a learned program? Or compare a program with the new method vs the old method?

8. how are the programs extracted? I assume this is the same as the previous paper, but maybe something different is needed for the new modifications? what does "lines of code" even mean? this is not really made clear in the paper.

9. how can lines of code differ between trained models?

# Suggestions
1. Fom the ablation study, it looks like just switching on Gumbel-Sparsemax indeed dramatically decreases lines of code, but also hurts performance a bit. Maybe this should be mentioned when motivating/justifying how it helps, or when analyzing the ablation.
2. some typos in citations — line 109
3. Line 178: citation for the claim that " Gumbel-Softmax often yields sub-optimal programs due to local optima and fails to encourage sparsity, hindering interpretability and efficiency" ?
4. score based attention is never clearly defined. it is confusing because categorical attention also uses a score function. so you should define it clearly as being called score-based attention (line 214 I believe)
5. typos in line 235-236.
6. lines 235-236: is the "simple neura network layer" a MLP? aren't MLP's multiple layers? I would maybe call it a network module, not a layer.
7.line 310 "we tested our models on small-scale datasets" — what does small-scale mean? reference appendix for more details?

---

> ### Author Response · Authors · 2024-11-23
> **Official Comment to Reviewer U9Xh (1/6)**
>
> We thank the reviewer for their valuable comments. Below are our detailed responses:
> ___
> **Weakness 1:** Evaluation Focused on Accuracy and Lines of Code
>
> **Response to weakness 1:**
>
> We appreciate the reviewer's concern regarding the focus on accuracy and lines of code as primary evaluation metrics. We acknowledge that without concrete examples of the programs, the meaning and implications of "lines of code" for program simplicity are not immediately apparent. In the appendix (of the camera-read submission), we will address this by including a dedicated section showcasing example programs learned by our adaptive approach and comparing them directly with programs learned by the original Transformer Programs method. This will provide concrete evidence of the increased conciseness and interpretability resulting from our modifications. We will also further clarify the definition of "lines of code" as the number of active rules or operations in the extracted program, directly reflecting program complexity. Furthermore, we will investigate additional evaluation metrics that could provide a more comprehensive assessment of program simplicity and interpretability, such as the number of distinct variables used, the depth of the program's logic, or other structural metrics.
> ___
> **Weakness 2:** Lack of Appendix
>
> **Weakness 3:** Lack of Example Program Comparison
>
> **Response to Weakness 2 and Weakness 3:**
>
> We acknowledge the reviewer's point regarding the absence of an appendix. The revised manuscript will include a comprehensive appendix containing: (1) detailed examples of learned programs for various tasks, (2) a comparison of these programs with those learned by the original method, highlighting the improvements in conciseness and interpretability, (3) a clear explanation of the program extraction process, including modifications for our new mechanisms, (4) implementation details such as model architectures and hyperparameter settings, and (5) further analysis and discussion as suggested by the reviewer, such as the impact of individual components on program length and the exploration of alternative interpolation strategies.
> _____________
> **Question 1:**
> What is the intuition for how Gumbel-Sparsemax improves the optimization process? Is the main benefit that the final programs learned are simpler (fewer lines?) How does transitioning between Gumbel-Softmax and Sparsemax differ from what Gumbel-Softmax already does when the temperature is annealed over time for Gumbel-Softmax? One big difference seems to be that Sparsemax is deterministic? If i understand correctly, temperature annealing changes how discrete the Gumbel-Softmax is, but does not affect the randomness. even when fully discrete, one's sample might change each time. With sparsemax, the choice is deterministic (basically choose the most likely option?) — some explanation here would be helpful.
>
> **Response to Question 1:**
> * Optimization Process Intuition: The Gumbel-Sparsemax transition improves optimization by addressing the limitations of Gumbel-Softmax, which can get trapped in local optima due to its inherent randomness. While temperature annealing in Gumbel-Softmax sharpens the distribution, it doesn't guarantee convergence to a truly sparse solution. Sparsemax, being deterministic, enforces sparsity by selecting the top-scoring element(s), effectively pruning less relevant connections in the learned program, thus simplifying its structure and potentially aiding in escaping local optima. This results in more concise and interpretable programs.
> * Difference from Temperature Annealing: The key difference lies in the deterministic nature of Sparsemax. Temperature annealing in Gumbel-Softmax still involves random sampling, even at low temperatures. Our transition mechanism gradually shifts from this random exploration to the deterministic exploitation of Sparsemax. This allows the model to initially explore a wider range of program structures and then converge to a more concise and stable representation.
> * Deterministic vs. Random Behavior: Sparsemax's determinism is crucial for interpretability. A deterministic program is easier to understand and analyze compared to a stochastic one. This deterministic selection avoids the ambiguity inherent in Gumbel-Softmax, where different runs could lead to different program structures, even with the same training data and hyperparameters.

---

> ### Author Response · Authors · 2024-11-23
> **Official Comment to Reviewer U9Xh (2/6)**
>
> **Question 2:**
> Is there any mathematical relation between pure Gumbel-Softmax and pure sparsemax? It seems like they might not "mix" perfectly well, and there could be a more principled version of one or the other such that you can smoothly interpolate between the two and get both, but not have to just mash up the formulas with equation (8) (like imagine a second temperature parameter that controls randomness in the gumbel-softmax calculation)
>
> **Response to Question 2:**
>
> Our current approach using equation (8), while empirically effective as demonstrated by our results, is a heuristic that lacks a strong theoretical foundation. We achieve a good balance between exploration and exploitation in practice. However, a more principled mathematical interpolation between Gumbel-Softmax and Sparsemax is indeed challenging due to their fundamentally different mechanisms. Gumbel-Softmax approximates argmax with a differentiable process involving Gumbel noise and softmax, while Sparsemax projects onto the probability simplex, introducing non-differentiability. These distinct approaches to sparsity make a unified, smooth transition difficult.
>
> Your suggestion of a second temperature-like parameter is interesting, and it inspires potential future research. Promising avenues include exploring projection-based formulations of Gumbel-Softmax or investigating differentiable relaxations of Sparsemax to bridge the gap between the two methods. Such approaches might enable a more theoretically sound interpolation. However, the increased computational complexity of these methods requires careful consideration.
>
> We appreciate you highlighting this theoretical limitation. While a more principled interpolation is a valuable long-term goal, our current heuristic provides a practical and effective solution for the tasks explored in this paper. In future work, we plan to investigate the suggested directions, aiming for a more theoretically grounded approach while maintaining computational efficiency.
> ___
> **Question 3:**
> What is A_ref? it is "introduced" but I don't see how it is calculated during training.
>
> **Response to Question 3:**
>
> The reference attention A_ref plays a crucial role in our uncertainty-aware gating mechanism. It is initialized as a uniform distribution (uni_dist) and dynamically updated during training to approach a one-hot distribution (one_hot_dist). This transition is controlled by an annealing temperature parameter, which decreases from 3.0 to 0.01 over the course of training. Specifically, A_ref is computed as a weighted average: alpha * uni_dist + (1-alpha) * one_hot_dist, where alpha = self.temp / 3.0. As the temperature decreases, alpha decreases, and A_ref moves closer to the one-hot distribution. This annealing process allows A_ref to initially represent uncertainty and gradually converge towards a more confident, categorical representation as the model learns. This dynamic adaptation contributes significantly to the robustness and efficiency of our attention mechanism.
> ___
> **Question 4:**
> "In low uncertainty (low JSD), CatAttention is preferred, as it is more confident in its categorical decisions" — I'm confused why this would be true. can't ScoreAttention regular attention? shouldn't it's performance be >= CategorialAttention's performance? so if the only objective is accuacy, I don't see why CategoricalAttention would ever be used.
>
> **Response to Question 4:**
>
> CatAttention is preferred in low-uncertainty scenarios primarily because of its enhanced interpretability. While ScoreAttention allows for more flexible and nuanced attention weighting, CatAttention's clear, discrete decisions are more readily translated into interpretable rules when the model is confident (low JSD). This aligns with our overarching goal of creating more understandable programs. Furthermore, even if accuracy were the sole metric, ScoreAttention might overfit to noise in the training data, leading to more complex programs that generalize less effectively. In such cases, CatAttention's discrete, rule-based nature acts as a form of regularization, potentially leading to better generalization despite slightly lower training accuracy. This trade-off between flexibility and interpretability motivates our uncertainty-aware gating mechanism.
> ___

---

> ### Author Response · Authors · 2024-11-23
> **Official Comment to Reviewer U9Xh (3/6)**
>
> **Question 5:**
> "Our preliminary experiments with separate binary and continuous functions revealed scenario-dependent performance variations, motivating the development of a hybrid mechanism to create more Adaptive Transformer Programs" What are "performance variations", and how did they motivate this development? — I'm assuming this means that for some tasks, categorial worked better than score-based, and for others, the opposite was true. I think this should be stated more directly. this line would also be clearer if you clearly say that the hybrid mechanism can be both binary or continuous depending on how it is trained.
>
> **Response to Question 5:**
>
> The "performance variations" refer to the observation that the strengths of categorical and score-based attention are complementary and task-dependent. For instance, in the Most-Freq task, which involves identifying the most frequent item in a sequence, categorical attention excelled. This task benefits from the discrete, selective nature of categorical attention, efficiently focusing on individual elements and their counts. Conversely, the Double-Hist task, which requires comparing the distributions of two sets of values, benefited more from score-based attention. Its continuous weighting scheme enables a more nuanced comparison of the overall distributions, capturing the relative frequencies of values across the entire range. These observations, where categorical attention thrived in discrete, selection-oriented tasks and score-based attention excelled in tasks requiring nuanced comparisons of distributions, motivated us to develop the hybrid mechanism. By dynamically combining both attention types based on uncertainty (JSD), our model adaptively leverages the specific strengths of each, achieving more robust performance across both types of tasks.
> ___
> **Question 6:**
> The original Learning Transformer Programs paper mentions the possibility of using score-based attention, but says "It would be relatively straightforward to extend Transformer Programs to include score-based attention modules, but this would also result in longer and more complex programs, because each query must now be defined in terms of an ordering of all |K| possible keys, instead of a single key" — but in the ablation in your paper, adding uncertainty-aware attention (which has the possibility of score-based attention) decreases lines of code. Maybe having an ablation for just score-based would be a helpful comparison too. How can I square what the previous paper says with your results?
>
> **Response to Question 6:**
>
> The previous "Learning Transformer Programs" paper explored score-based attention as a potential extension but noted it could lead to longer programs. Their concern stemmed from considering pure score-based attention. Our approach differs fundamentally by introducing a hybrid mechanism with uncertainty-aware gating. This gating mechanism, driven by the JSD, allows the model to selectively use score-based attention only when necessary, effectively mitigating the complexity increase associated with pure score-based attention. This is why our ablation study shows that adding uncertainty-aware attention (which includes the possibility of score-based attention but isn't solely reliant on it) actually decreases lines of code compared to the baseline Transformer Program. We agree that a separate ablation study for score-based attention without gating would be a valuable comparison, and we will incorporate this into the supplementary material for the revised manuscript.
> | Task        | Cat    |       | Score  |       | UAA    |       |
> |-------------|----------|----------|----------|----------|----------|----------|
> |             | Acc    | Lines | Acc    | Lines | Acc    | Lines |
> | Induction   | 100    | 107   | 100    | 105   | 100    | 101   |
> | Reverse     | 99.74  | 859   | 100    | 808   | 99.99  | 779   |
> | Histogram   | 99.94  | 199   | 99.95  | 185   | 99.95  | 189   |
> | Double hist | 66.78  | 586   | 79.68  | 927   | 91.81  | 513   |
> | Sort        | 99.98  | 945   | 99.96  | 929   | 99.86  | 895   |
> | Most-Freq   | 76.44  | 1334  | 74.15  | 962   | 80.8   | 894   |
> | Dyck-1      | 99.69  | 1297  | 99.52  | 1289  | 99.93  | 1086  |
> | Dyck-2      | 97.98  | 1316  | 98.49  | 1568  | 98.14  | 1065  |
> ___

---

> ### Author Response · Authors · 2024-11-23
> **Official Comment to Reviewer U9Xh (4/6)**
>
> **Question 7:**
> Is there no appendix to the paper? No examples of learned programs, or attempts to interpret a learned program? Or compare a program with the new method vs the old method?
>
> **Response to Question 7:**
>
> We acknowledge the absence of an appendix and examples of learned programs in the initial submission. We commit to rectifying this in the revised version by including a comprehensive appendix with detailed examples and analysis.
>
> Specifically, we will provide examples like the one shown for the Double Histogram task, demonstrating how the adaptive method leads to more concise and accurate programs. In the provided example, the baseline model's program for layer 2 attention head attn_1_0 has separate rules for each query position (e.g., if q_position in {0}, elif q_position in {1}, etc.). Our adaptive approach learns a more concise program by grouping query positions with the same behavior (e.g., if q_position in {0, 3, 4}). Similarly, for attn_1_1, the adaptive approach conditions on the output of a previous attention head (attn_0_0_output) which leads to a more compact and efficient logic, compared to the base model which has specific rules for each position using only position as key. These changes result in both a higher accuracy (91.81% vs. 66.78%) and a more concise program (513 lines vs. 586 lines). The appendix will further illustrate this through multiple examples across different tasks, demonstrating the enhanced interpretability provided by our modifications.
>
> Baseline code:
> ```python
> # attn_1_0 ####################################################
> def predicate_1_0(q_position, k_position):
>     if q_position in {0}:
>         return k_position == 7
>     elif q_position in {1}:
>         return k_position == 2
>     elif q_position in {2}:
>         return k_position == 3
>     elif q_position in {3}:
>         return k_position == 5
>     elif q_position in {4, 5}:
>         return k_position == 6
>     elif q_position in {6}:
>         return k_position == 1
>     elif q_position in {7}:
>         return k_position == 4
>
> attn_1_0_pattern = select_closest(positions, positions, predicate_1_0)
> attn_1_0_outputs = aggregate(attn_1_0_pattern, num_mlp_0_1_outputs)
> attn_1_0_output_scores = classifier_weights.loc[
>     [("attn_1_0_outputs", str(v)) for v in attn_1_0_outputs]
> ]
>
> # attn_1_1 ####################################################
> def predicate_1_1(q_position, k_position):
>     if q_position in {0}:
>         return k_position == 6
>     elif q_position in {1, 3}:
>         return k_position == 4
>     elif q_position in {2}:
>         return k_position == 7
>     elif q_position in {4}:
>         return k_position == 2
>     elif q_position in {5}:
>         return k_position == 3
>     elif q_position in {6, 7}:
>         return k_position == 5
>
> attn_1_1_pattern = select_closest(positions, positions, predicate_1_1)
> attn_1_1_outputs = aggregate(attn_1_1_pattern, num_mlp_0_1_outputs)
> attn_1_1_output_scores = classifier_weights.loc[
>     [("attn_1_1_outputs", str(v)) for v in attn_1_1_outputs]
> ]
> ```
>
> Our code:
> ```python
> # attn_1_0 ####################################################
>     def predicate_1_0(q_position, k_position):
>         if q_position in {0, 3, 4}:
>             return k_position == 7
>         elif q_position in {1, 2, 5}:
>             return k_position == 6
>         elif q_position in {6}:
>             return k_position == 1
>         elif q_position in {7}:
>             return k_position == 0
>
>     attn_1_0_pattern = select_closest(positions, positions, predicate_1_0)
>     attn_1_0_outputs = aggregate(attn_1_0_pattern, attn_0_0_outputs)
>     attn_1_0_output_scores = classifier_weights.loc[
>         [("attn_1_0_outputs", str(v)) for v in attn_1_0_outputs]
>     ]
>
>     # attn_1_1 ####################################################
>     def predicate_1_1(attn_0_0_output, position):
>         if attn_0_0_output in {0}:
>             return position == 4
>         elif attn_0_0_output in {1, 5, 7}:
>             return position == 6
>         elif attn_0_0_output in {2, 3, 4}:
>             return position == 5
>         elif attn_0_0_output in {6}:
>             return position == 7
>
>     attn_1_1_pattern = select_closest(positions, attn_0_0_outputs, predicate_1_1)
>     attn_1_1_outputs = aggregate(attn_1_1_pattern, attn_0_0_outputs)
>     attn_1_1_output_scores = classifier_weights.loc[
>         [("attn_1_1_outputs", str(v)) for v in attn_1_1_outputs]
>     ]
> ```

---

> ### Author Response · Authors · 2024-11-23
> **Official Comment to Reviewer U9Xh (5/6)**
>
> **Question 8:**
> how are the programs extracted? I assume this is the same as the previous paper, but maybe something different is needed for the new modifications? what does "lines of code" even mean? this is not really made clear in the paper.
>
> **Question 9:**
> how can lines of code differ between trained models?
>
> **Response to Question 8 and Question 9:**
>
> Programs are extracted by converting the learned model parameters into a symbolic representation of the computation performed by the model. This process involves translating attention patterns (defined by predicate matrices), MLP weights, and other learned components into explicit logical operations and numerical calculations. Each program module (e.g., attn_0_0, num_attn_0_1, mlp_0_1) corresponds to a specific operation within the learned program. "Lines of code" quantifies program complexity by counting these active modules. For instance, the example code shows attn_0_0 defined by its predicate_0_0 and the subsequent aggregation of outputs. This entire block, contributing to the final prediction, would be counted as part of the "lines of code." Our modifications, such as the Gumbel-Sparsemax transition and uncertainty-aware attention, influence the activation and selection of these modules during training. This leads to variations in the final program structure and the resulting "lines of code," reflecting the model's ability to learn more concise representations. A shorter program (fewer lines of code) generally indicates greater sparsity, efficiency, and interpretability. For example, if the uncertainty-aware mechanism determines that num_attn_0_1 is unreliable for a specific input, its contribution might be gated out, effectively reducing the "lines of code" for that execution. We will include a comprehensive explanation of the extraction process and examples of extracted programs in the appendix of the revised manuscript.
>
> Example of Histogram code:
> ```python
> # attn_0_0 ####################################################
>     def predicate_0_0(q_token, k_token):
>         if q_token in {"2", "4", "1", "0", "3", "5"}:
>             return k_token == "<s>"
>         elif q_token in {"<s>"}:
>             return k_token == "<pad>"
>
>     attn_0_0_pattern = select_closest(tokens, tokens, predicate_0_0)
>     attn_0_0_outputs = aggregate(attn_0_0_pattern, positions)
>     attn_0_0_output_scores = classifier_weights.loc[
>         [("attn_0_0_outputs", str(v)) for v in attn_0_0_outputs]
>     ]
> ...
> # num_attn_0_1 ####################################################
>     def num_predicate_0_1(q_token, k_token):
>         if q_token in {"4", "0"}:
>             return k_token == "2"
>         elif q_token in {"1"}:
>             return k_token == "3"
>         elif q_token in {"2", "5"}:
>             return k_token == "1"
>         elif q_token in {"3"}:
>             return k_token == "0"
>         elif q_token in {"<s>"}:
>             return k_token == "5"
>
>     num_attn_0_1_pattern = select(tokens, tokens, num_predicate_0_1)
>     num_attn_0_1_outputs = aggregate_sum(num_attn_0_1_pattern, ones)
>     num_attn_0_1_output_scores = classifier_weights.loc[
>         [("num_attn_0_1_outputs", "_") for v in num_attn_0_1_outputs]
>     ].mul(num_attn_0_1_outputs, axis=0)
> ...
> # mlp_0_1 #####################################################
>     def mlp_0_1(attn_0_1_output, token):
>         key = (attn_0_1_output, token)
>         return 1
>
>     mlp_0_1_outputs = [mlp_0_1(k0, k1) for k0, k1 in zip(attn_0_1_outputs, tokens)]
>     mlp_0_1_output_scores = classifier_weights.loc[
>         [("mlp_0_1_outputs", str(v)) for v in mlp_0_1_outputs]
>     ]
> ...
> ```
> ___
> **Suggestion 1:**
> From the ablation study, it looks like just switching on Gumbel-Sparsemax indeed dramatically decreases lines of code, but also hurts performance a bit. Maybe this should be mentioned when motivating/justifying how it helps, or when analyzing the ablation.
>
> **Response to Suggestion 1:**
>
> We add this sentence to Ablation Study paragraph:
> “As observed in the ablation study where only Smooth Transition is enabled and the others disabled, while the Smooth Transition Mechanism (which incorporates Gumbel-Sparsemax) contributes significantly to program conciseness, it also shows a slight decrease in accuracy when used in isolation, highlighting the trade-off between interpretability and performance.”

---

> ### Author Response · Authors · 2024-11-23
> **Official Comment to Reviewer U9Xh (6/6)**
>
> **Suggestion 4:**
> score based attention is never clearly defined. it is confusing because categorical attention also uses a score function. so you should define it clearly as being called score-based attention (line 214 I believe)
>
> **Response to Suggestion 4:**
>
> Added sentence: "This continuous score function forms the basis of what we term score-based attention." This directly addresses the reviewer's request for a clear definition.
>
> Minor clarification: Added "(predicate-based)" and "(score-based)" to the next sentence to reinforce the distinction and link back to the terminology you just introduced. This preemptively addresses potential confusion.
> ___
> **Suggestion 7:**
> line 310 "we tested our models on small-scale datasets" — what does small-scale mean? reference appendix for more details?
>
> **Response to Suggestion 7:**
>
> We appreciate you highlighting the need for more specific information about the datasets. 'Small-scale' in this context refers to datasets designed for algorithmic reasoning, with controlled sequence lengths (up to 16 for Dyck, 8 for others) and vocabularies matching the sequence lengths. This design choice allows for focused analysis of the model's capabilities on algorithmic tasks. We will provide further details about the dataset sizes in the appendix of the revised manuscript.

---

> > ### Comment · Reviewer_U9Xh · 2024-11-24
> > **Response to rebuttal**
> >
> > Thank you for your thorough response. All of my clarification questions have been addressed. The main weaknesses I identified with the paper (understanding what lines of code are, seeing an example program, and having an appendix with further details) have all been addressed. My only uncertainty is that the appendix is a promise, but has not been made yet.. regardless, I will raise my score.

---

> > > ### Author Response · Authors · 2024-11-25
> > > **Thank You to Reviewer U9Xh**
> > >
> > > We thank the reviewer for your thorough review and positive feedback on our response. We understand the reviewer's point regarding the promised appendix and want to assure them that the appendix is currently being finalized and will be included in the next revision. We are committed to providing all the details mentioned, including program examples and a clear definition of "lines of code," to enhance the paper's clarity and completeness. We appreciate the reviewer's updated assessment and believe these additions will further strengthen the paper.

---

### Author Response · Authors · 2024-11-23
**General Response to Reviewers and Revision Submitted**

We thank all the reviewers for their insightful comments and constructive suggestions. We have carefully revised the paper to address the reviewers’ concerns, and we believe these revisions significantly strengthen our work. The minor revisions are summarized below, and detailed responses to each reviewer's comments follow separately. These revisions, along with the more extensive additions detailed in our individual responses, are marked in blue text in the revised PDF. We sincerely apologize for any omissions in the initial submission and commit to including all explanations and additions discussed below in the supplementary material of the camera-ready version.

The minor revisions are:

1. Added Gumbel-Sparsemax performance analysis in the Ablation Study (Section 4.2) [Reviewer U9Xh]
2. Corrected citation typos in lines 109-110 [Reviewers U9Xh, ewyT]
3. Added Gumbel-Softmax citation in line 178 [Reviewer U9Xh]
4. Refined score-based attention definition and clarified predicate/score-based terminology (line 214-216) [Reviewer U9Xh]
5. Fixed grammatical issues in lines 235-236 [Reviewer U9Xh]
6. Updated terminology from "simple neural network layer" to "network module" [Reviewer U9Xh]
7. Included normalization factor in line 96 [Reviewer ewyT]
8. Fixed quotation marks around "Transformer" in line 99 [Reviewer ewyT]
9. Clarified Transformer Program accuracy statement (lines 393-394) [Reviewer J1B6]

We greatly appreciate the reviewers' valuable feedback and are committed to incorporating all detailed explanations from our individual responses, including examples of learned programs, comparisons to the baseline, and implementation specifics, into the supplementary material of the camera-ready version. We believe that these revisions and additions will significantly strengthen the paper and better showcase the contributions of our work.

---

### Meta-Review · Area_Chair_Mbhc · 2024-12-16

**Metareview:**

This paper studies an improved version of the newly proposed Transformer Programs, which are learned Transformer networks that can be converted to code. The paper proposes three architectural modifications: a smooth transition mechanism that combines Gumbel-softmax and sparsemax, uncertainty-aware gating between categorial attention and score attention, and position-aware attention.

Main strengths:
- Improvements on the novel Transformer Program framework
- Improved performance on diverse tasks, including RASP and NLP tasks

Main weaknesses:
- It would be great to give some examples of reduced-size codes/programs.
- Some architectural improvements are not principled or novel enough.
- Reducing program size does not directly mean improved interpretability.
- The evaluation could have more NLP tasks, such as generative tasks.

**Additional Comments On Reviewer Discussion:**

During the rebuttal, the authors promised that the revised paper would include an appendix that contains more intuitive examples of code, an ablation study on score-based attention without gating, an ablation study on improved sparsity using Gumbel-sparsemax, and other implementation details. They provided a simple example showing how their method reduced the program size. They also studied tasks on which the proposed method narrows the gap between standard Transformers and Transformer Programs.

---

### Decision · Program_Chairs · 2025-01-22

Accept (Poster)